# Uncertainty analysis of a rainfall threshold estimate for stony debris flow based on the Backward Dynamical Approach

Marta Martinengo[1], Daniel Zugliani[1], and Giorgio Rosatti[1]

[1]Department of Civil, Environmental and Mechanical Engineering, University of Trento, Trento, Italy

**Abstract.** A rainfall threshold is a function of some characteristic rainfall quantities that provides the conditions beyond which the probability of debris-flow occurrence is considered significant. Many uncertainties may affect the thresholds calibration and, consequently, its robustness. This study aims to assess the uncertainty in the estimate of a rainfall threshold for stony debris flow based on the Back Dynamical Approach, an innovative method to compute the rainfall duration and averaged intensity strictly related to a measured debris flow. The uncertainty analysis is computed by performing two Monte Carlo cascade simulations: (i) to assess the variability in the event characteristics estimate due to the uncertainty in the Back Dynamical Approach parameters and data and (ii) to quantify the impact of this variability on the threshold calibration. This procedure applied to a case study highlights that the variability in the event characteristics can be both low and high. Instead, the threshold coefficients have a low dispersion showing good robustness of the threshold estimate. Moreover, the results suggest that some event features are correlated with the variability of the rainfall event duration and intensity. The proposed method to assess the uncertainty is suitable to be also applied to other threshold calibration approach.

## 1 Introduction

In mountain regions, rainfall-induced natural phenomena, as shallow landslides and debris flows, are relatively frequent events that have a significant impact on the territory in which they occur, causing damages and, in some cases, casualties (Fuchs et al., 2013; Dowling and Santi, 2014; Cánovas et al., 2016). The risk management of these phenomena is crucial to reduce their effects on the territory and it is based on both active and passive mitigation strategies. An early warning system is an example of a passive mitigation tool (Huebl and Fiebiger, 2005) as it allows to activate prevention measures (e.g. evacuation sets out in the civil protection plans) before the expected event occurs.

The early warning systems for these phenomena are mainly based on rainfall thresholds (Chien-Yuan et al., 2005; Segoni et al., 2018), namely rainfall conditions beyond which the occurrence probability of a rainfall-induced event is considered significant. In this framework, most rainfall thresholds are power law relations expressing the rainfall event cumulated or intensity as a function of the event duration (Segoni et al., 2018). A considerable literature deals with this topic (e.g. Caine, 1980; Guzzetti et al., 2008; Winter et al., 2010; Jakob et al., 2012; Staley et al., 2013; Marra et al., 2014; Zhou and Tang, 2014; Iadanza et al., 2016; Marra et al., 2016; Pan et al., 2018).

In some studies rainfall thresholds concern a wide typology of phenomena (Segoni et al., 2018), other works focus on both shallow landslides and debris flows (e.g. Baum and Godt, 2010; Cepeda et al., 2010), other on shallow landslides (e.g.

Giannecchini, 2005; Frattini et al., 2009) and finally some studies are specifically conceived for debris flow (e.g. Nikolopoulos et al., 2014; Giannecchini et al., 2016; Li et al., 2016).

Power-law thresholds can be derived in the following way. Given a historical dataset of rainfall-induced events, the rainfall associated with each event is determined and described in terms of the couple of synthetic quantities employed in the threshold (e.g. rainfall event cumulated - event duration). Classically, these quantities are defined only on the basis of a hyetograph analysis (Segoni et al., 2018), without considering the characteristics of the rainfall-induced phenomenon. In a log-log plane, the resulting set of couples becomes a cloud of points and the power-law function is a straight line. Starting from these couples set, the threshold is determined by locating the straight line in the log-log plane using one of the several estimate strategies available in the literature, e.g. manual methods, statistical approaches, probabilistic procedures (Guzzetti et al., 2007; Segoni et al., 2018). The result is the calibrated rainfall threshold.

One of the critical issues of the calibration is the uncertainty related to both data and models parameters (Gariano et al., 2020). Here with the term "model", we indicate generically a single equation or a set of operations that, given some input data and model parameters, provide an output. In the case of the rainfall threshold, the uncertainties derive mainly from direct data error measurements (e.g. in rainfall), from the non-unique definition of the models parameters (e.g. distance within which to select the rain gauge to define the event precipitation) and from the strategy used to calibrate the threshold. The result is an uncertainty framework that can significantly impact the threshold estimate.

Some studies have already investigated the uncertainty in threshold determination, focusing on some aspects that can affect the hyetograph or the event synthetic quantities used in the threshold. For instance, Nikolopoulos et al. (2014) has analysed the consequence of the spatial variability of the precipitation while Marra (2019) and Gariano et al. (2020) have investigated the effect of the rainfall temporal resolution. Moreover, the uncertainty arising from the choice of the reference rain gauge and the differences between the radar and the rain gauge measurements have been examined in Rossi et al. (2017). Besides, the effect of the uncertainty in triggering rainfall estimate has been investigated in Peres et al. (2018) while Abraham et al. (2020) has analysed the consequences of the scale of analysis, the rain gauge selection and how the intensity is quantified.

Rosatti et al. (2019) has introduced an innovative method to calibrate an intensity-duration rainfall threshold for stony debris flow, a particular type of debris flow, frequent in some mountain areas as in the Alps, in which the presence of silt and/or clay in the mixture is negligible and the internal stresses are mainly caused by the collision among the particles (e.g. Takahashi, 2009; Stancanelli et al., 2015; Bernard et al., 2019). The new method, called Backward Dynamical Approach (BDA), starts from the knowledge of the volume of sediments deposited after an event and, thanks to a schematic description of the stony debris-flow dynamic, it is able to identify, in the related hyetograph, the rainfall event volume, intensity and duration, strictly pertaining to the debris-flow event. Hence, the BDA differs from the classical literature approaches since the synthetic quantities describing the rainfall events are defined involving not only the forcing (i.e. the hyetograph) but also the dynamic of the rainfall-induced event.

This work focuses on the uncertainty deriving from data and parameters inherent to the BDA, leaving out the uncertainty related to the hyetograph, already investigated in the literature. In particular, the aim is to perform an uncertainty analysis on the threshold calibration to check the robustness of the BDA. To reach the goal, among the different strategies and methods

available in the literature (e.g. Helton et al., 2006; Coleman and Steele, 2018; Hofer, 2018), we have chosen the Monte Carlo (MC) approach. With this tool, we have developed a proper methodology composed of two MC cascade simulations and we have applied it to a dataset concerning a specific study area. Detailed analysis of intermediate and final results have also been performed to better understand the uncertainty analysis outcomes.

The paper structure is the following. A brief description of the BDA method is presented in Sect. 2. The study area and data are described in Sect. 3. The method used to assess the uncertainty propagation in the BDA-based threshold calibration is described in Sect. 4. The obtained results are presented and discussed in Sect. 5. Conclusions end the paper.

## 2 The BDA-based threshold calibration

As mentioned in the Introduction, the BDA determines the rainfall event intensity and duration, namely the couple $(I, D)$, associated to a stony debris flow by using not only the hyetograph but also information concerning the occurred debris flow.

Starting from the knowledge of the deposited volume $V_{dep}$ occupied by the sediments after a debris-flow event and thanks to a simplified global volumetric description of the debris-flow dynamic (Fig.1), the rainfall volume pertaining to the debris flow $V_r^{DF}$, defined as the volume of water necessary to convey downstream $V_{dep}$ as a mixture, can be express as:

$$V_r^{DF} = \frac{c_b - c}{c} V_{dep} \tag{1}$$

where $c_b$ is the concentration of the sediment in the bed, constant and assumed equal to $0.65$ (Takahashi, 2014), and $c$ is a reference volumetric solid concentration of the given debris flow.

The expression of Takahashi (1978), valid in permanent and uniform conditions, can be used as reference concentration:

$$c = \min\left(\frac{i_f}{\Delta(\tan\psi - i_f)}, 0.9 c_b\right) \tag{2}$$

where $i_f$ is the bed slope, $\psi$ is the dynamic friction angle of the sediments and $\Delta = (\rho^s - \rho^l)/\rho^l$ is the sediment relative submerged density, constant and assumed equal to $1.65$ (e.g. Prancevic and Lamb, 2015), where $\rho^l$ and $\rho^s$ are, respectively, the liquid and solid constant density. According to the assumptions of the BDA, the reference concentration is evaluated considering the bed slope in the last portion of the debris-flow channel, just upstream of the deposition area. This means that information concerning the triggering conditions and the detailed evolution of the debris flow in the upper part of the basin are not considered.

The rainfall volume pertaining to the debris flow can also be expressed as the product of the rainfall volume per unit area $E$ and the event basin area $A_b$:

$$V_r^{DF} = E A_b \tag{3}$$

from which, a backward dynamical expression for the rainfall volume per unit area can be obtained by equating (1) with (3):

$$E = \frac{1}{A_b}\frac{c_b - c}{c} V_{dep} \tag{4}$$

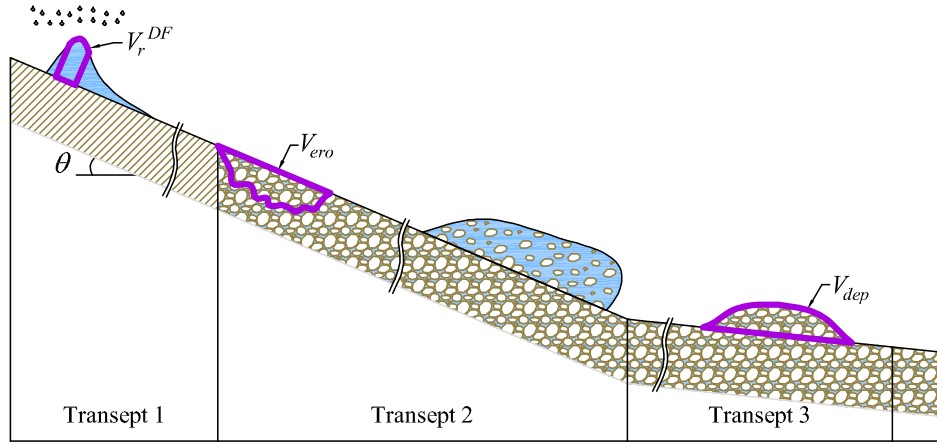

**Figure 1.** Conceptual Lagrangian volumetric description of debris-flow dynamic from Rosatti et al. (2019). The scheme is divided into three transepts: transept 1 is characterised by the runoff formation; the bed material erosion and the achievement of equilibrium conditions occur in transept 2; transept 3 is characterised by the deposition of sediments with water entrapment. $V_r^{DF}$ is the rain volume pertaining to the debris flow, $V_{ero}$ is the bed volume variation related to the erosion, $V_{dep}$ is the deposited volume occupied by the sediments and $\theta = \arctan(i_f)$ is the inclination angle of the bed with respect to a reference horizontal direction.

On the other hand, $E$ can be obtained from the forcing of the phenomenon, namely the hyetograph. Under the assumption of uniform rainfall over the basin, the hydrological expression for $E$ is:

$$E = \int_{t_1}^{t_2} i(t)\,dt \tag{5}$$

where $i(t)$ is the measured rainfall intensity and $t_1$ and $t_2$ are the unknown start and end times related to the debris-flow duration. In the absence of event detailed data, these times are expressed as:

$$\begin{cases} t_1 = t_{max} - \Delta t_1 \\ t_2 = t_{max} + \Delta t_2 \end{cases} \tag{6}$$

where $t_{max}$ is the instant of maximum intensity during the event and $\Delta t_1$ and $\Delta t_2$ are unknown intervals. These intervals can be obtained equating the right hand side terms of Eq. (5) and (4):

$$\int_{t_{max}-\Delta t_1}^{t_{max}+\Delta t_2} i(t)\,dt = \frac{1}{A_b}\frac{c_b - c}{c} V_{dep} \tag{7}$$

Because of the measurement technique, $i(t)$ is a piecewise constant function on time intervals $\delta t$, namely $i(k)$. Consequently, reference times becomes: $t = k\delta t$, $t_{max} = M\delta t$, $\Delta t_1 = n_1 \delta t$ and $\Delta t_2 = n_2 \delta t$ where $M$ is the number of time intervals that identifies the peak and now $n_1$ and $n_2$ are unknown integers. Another consequence is that the integral in Eq. (7) must be rewritten in discrete form (namely a summation) and that the previous equation cannot be satisfied exactly.

An approximated algorithm, able to determine in an univocal way the unknowns, can be introduced: starting from zero and increasing of one unit alternatively $n_1$ and $n_2$, the first couple $\hat{n}_1, \hat{n}_2$ such that

$$\sum_{k=M-\hat{n}_1}^{M+\hat{n}_2} i(k)\,\delta t \geq \frac{1}{A_b}\frac{c_b - c}{c}V_{dep} \tag{8}$$

is the searched couple. If a zero-intensity interval is reached, the sum stops being symmetrical with respect to $M$ and only either $n_1$ or $n_2$ is increased until the previous relation is satisfied.

Finally, the duration $D$ and the average intensity $I$, can be expressed as:

$$D = \Delta t_1 + \Delta t_2 = (\hat{n}_1 + \hat{n}_2)\,\delta t \tag{9}$$

$$I = \frac{\sum_{k=M-\hat{n}_1}^{M+\hat{n}_2} i(k)\,\delta t}{D} \tag{10}$$

Once the $(I, D)$ couple is computed for each event of the available dataset, the rainfall threshold is estimated by using the frequentist method (e.g. Brunetti et al., 2010; Peruccacci et al., 2012). According to this method, the $(I, D)$ couples are plotted in a log-log $ID$ plane and a straight line fitting these points is determined. The slope and the intercept of this straight line are the logarithms of the coefficient of the following power law:

$$I = \hat{a}D^{-b} \tag{11}$$

The rainfall threshold is then obtained translating vertically the straight line in the log-log $ID$ plane so that the non-exceedance probability of the dataset events (namely the occurrence probability of debris flows related to $(I, D)$ points located below the threshold) is equal to a given value. The final expression is:

$$I = aD^{-b} \tag{12}$$

in which $a < \hat{a}$.

For more details on the BDA and the frequentist method, we refer the reader to the above-mentioned references.

## 3  Study area and data

The study area and data used in this analysis are the same as those used in Rosatti et al. (2019). In particular, the study area is the Trentino-Alto Adige/Südtirol region, in the north east of the Italian Alps (Fig. 2(a)). The region covers 13607 km$^2$, has an altitude range between 40 and 3900 m a.s.l. with mean about 1600 m a.s.l. (Fig. 2(b)) and a climate characterised mostly by a continental regime (Bisci et al., 2004; Nikolopoulos et al., 2014).

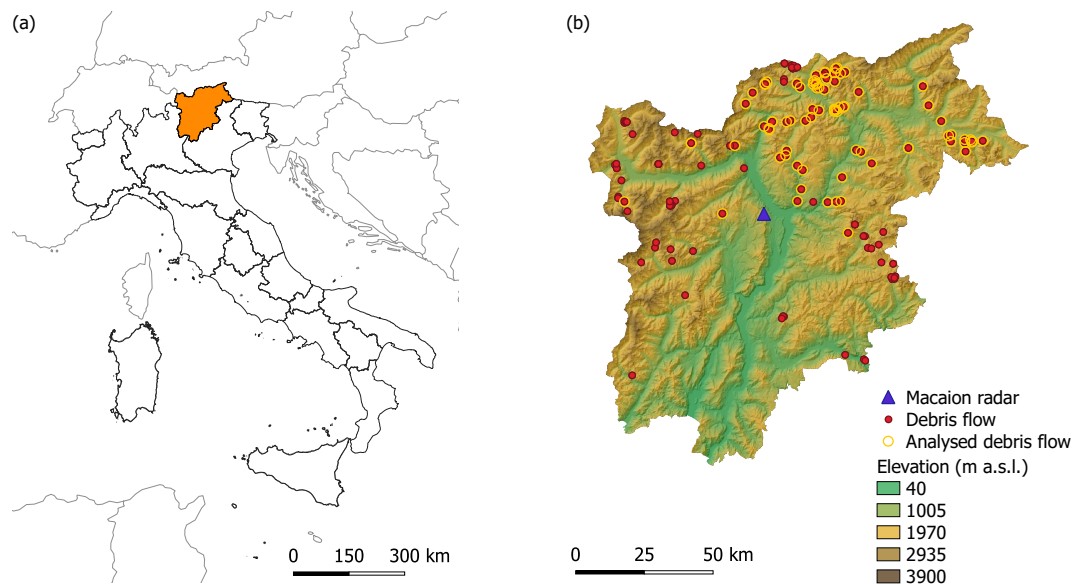

**Figure 2.** (a) Location of Trentino-Alto Adige/Südtirol region (Italy) and (b) the Macaion radar and debris-flow events: red dots show all debris flows while yellow circles highlight the suitable ones for the study.

The regional agencies between 2006 and 2016 have reported 161 debris flows (Fig. 2(b)) but only 139 events present the survey of the deposits, whose volumes range between 100 m$^3$ and 50000 m$^3$. In every event, sediment are characterised by the absence or, at least, the negligible presence of silt and clay thus resulting as stony debris flows.

The rainfall data associated to these events derives from a radar located in a central position with respect to the region, on the Mt. Macaion at 1866 m a.s.l. (Fig. 2(b)). A C-band Doppler weather radar measures the reflectivity $Z$ over an area of 120 km of radius and the rainfall is computed converting $Z$ into precipitation intensity $I$ (e.g. Uijlenhoet, 2001). Since radar data in mountain regions are typically affected by the beam shielding (Germann et al., 2006) that can cause errors in the measurements, the debris-flow events located in an area with a weakening of the signal greater than 90% are excluded from the dataset. Overall, the debris flows suitable for the analysis were 84 and are highlighted in Fig. 2(b) with circles.

Additional data required for the BDA, namely $i_f$, $A_b$, $\psi$ and $i(t)$, was defined for each event in the following way. The basin outlet has been located downstream of a segment with a sufficiently constant slope just upstream of the deposition area and the upstream basin area was determined. Then, $i_f$ has been calculated as the mean slope of the last 50 m of the torrent upstream of the outlet point. Besides, due to the scarcity of sediments information, $\psi$ was assumed to be equal to $35°$ for all the events. The hyetograph $i(t)$ was computed at each instant averaging over the respective basin area the radar intensities. In this way, both the spatial and temporal variability of the rainfall is taken into account.

Starting from this data and setting the non-exceedance probability equal to 5%, Rosatti et al. (2019) obtained the following threshold:

$$I = 6.2D^{-0.67} \tag{13}$$

From now on, the quantities involved in the calibration performed by Rosatti et al. (2019) will be considered as reference values and they will be indicated with a subscript $r$.

 # 4 Method

As described in Sect. 2, the BDA-based threshold calibration starts from the definition of the following input parameters and data for each considered event: $i_f$, $A_b$, $V_{dep}$, $\psi$ and $i(t)$. Subsequently, based on these values, what we call the "event characteristics" are computed for each analysed debris flow: first $c$ (Eq. (2)) and $E$ (Eq. (4)), and then $D$ (Eq. (9)) and $I$ (Eq. (10)). Finally, the $(I, D)$ couples of the events are used to calibrate the threshold, namely to quantify the threshold coefficients $a$ and $b$ of Eq. (12).

Coherently to the estimate procedure, the uncertainty analysis of the BDA-based threshold calibration is divided into three parts (Fig. 3). First, the uncertainty characterisation of the input parameters and data is determined (Fig. 3(a)). Then, for each debris flow, the uncertainty analysis of the event characteristics is performed with an MC simulation, starting from the uncertain quantities above defined (Fig. 3(b)). Finally, a further MC simulation is carried out to perform the uncertainty analysis of the threshold, using as input the $(I, D)$ couples of the events obtained from the first MC simulation (Fig. 3(c)). In this way, the impact of the uncertain parameters and data on the threshold is quantified.

All the analyses are performed using the R software (R Core Team, 2013) and, in the following sections, we present the details of each part.

## 4.1 Uncertainty characterisation

As explained in the Introduction, in this study, the focus is on the uncertainty in the physical and morphological parameters and data used in the BDA to describe in a simplified way the debris-flow dynamic. Therefore, in this analysis, the variables considered are $i_f$, $A_b$, $V_{dep}$ and $\psi$. According to their estimate described in Sect. 3, these variables are mainly affected by epistemic uncertainty due to measurement and estimate errors and lack of information (Oberkampf et al., 2004).

The characterization of the uncertainty in the variable, namely the probability distribution function (pdf) of their values, has to be defined both in term of distribution type and statistical quantities (e.g. mean and variation coefficient $CV$) (Fig. 3(a)). Lacking certain data concerning the pdfs, according to Marino et al. (2008), all the variable are assumed to be uniformly distributed and, for each event, the means of the distributions are set equal to the corresponding reference values.

Regarding the deviations from the means, $\psi$ is the only variable whose variability is constrained by a validity range: for stony debris flow, according to Lane (1953) and Blijenberg (1995), $\psi$ can vary between $32°$ and $38°$. Assuming $35°$ as the mean of the $\psi$ distribution, the variability range $(32°, 38°)$ can be obtained by imposing $CV$ equals to about $5\%$.

The uncertainty in $V_{dep}$ can not be accurately estimated since the survey methodology, and the related measurement errors, used by regional agencies, is not univocal (Marchi et al., 2019). However, Brardinoni et al. (2012) has proposed for a similar study area a relative error of $10\%$ in the estimate of $V_{dep}$, namely a corresponding $CV$ equals to about $5\%$. Therefore, we assume this uncertainty value is valid for this analysis.

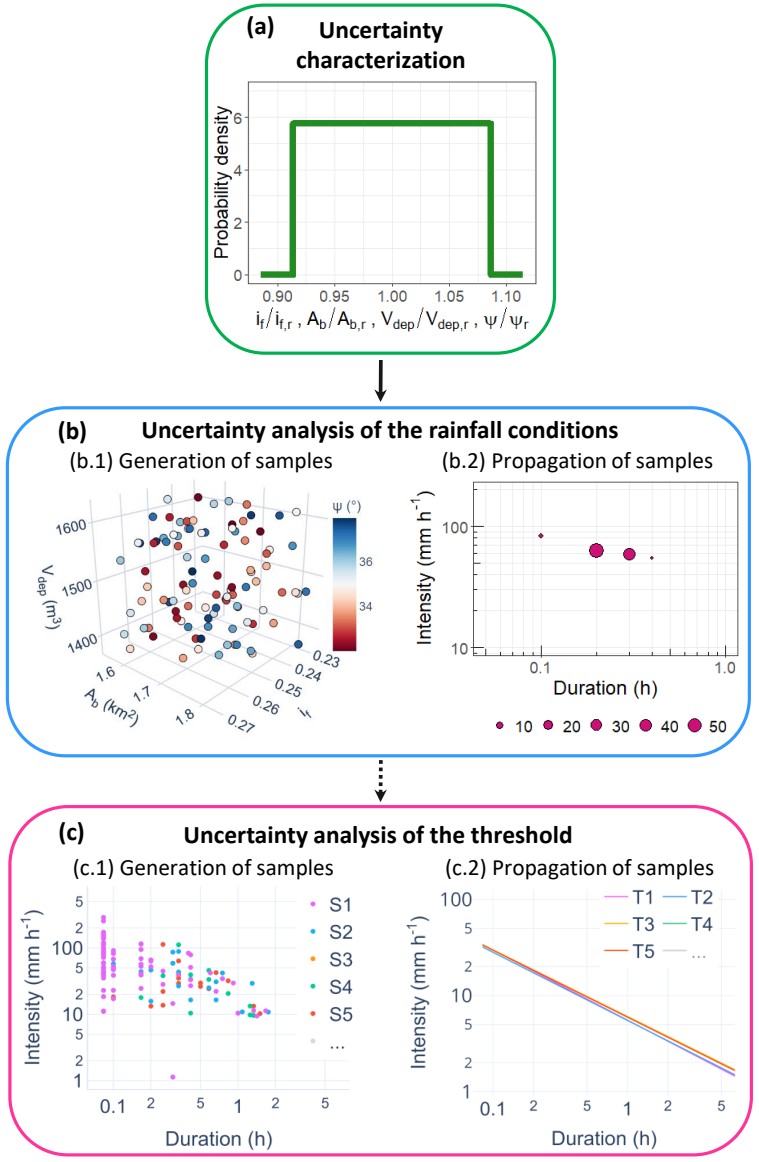

**Figure 3.** Scheme of the uncertainty analysis performed with two cascade MC simulations. (a) Uncertainty characterization of the parameters and data: the non-dimensional form of the uncertain parameters and data, obtained by dividing the variables by the reference values, are assumed to be uniformly distributed. (b) First MC simulation to compute the uncertainty analysis of the event characteristics for each debris flow: (b.1) samples generation performing the Latin Hypercube Sampling (LHS) and (b.2) propagation of samples to compute the event characteristics. The dots size in the log-log $ID$ plane indicates the absolute frequency of obtaining the $(I, D)$ couples. (c) Second MC simulation to perform the uncertainty analysis of the threshold: (c.1) random samples $S$ generation (one of the previous obtained $(I, D)$ couples for each event) and (c.2) propagation of samples to estimate the thresholds $T$.

**Table 1.** Probability distributions of the uncertain variables for each event. $i_{f,r}$, $A_{b,r}$ and $V_{dep,r}$ are the event reference values of the average slope, the basin area and the deposited sediments respectively.

| Variable | Probability distribution function |
|---|---|
| $i_f$ | $\text{Uniform}(i_{f,r}\,(1-5\%\sqrt{3}),\, i_{f,r}\,(1+5\%\sqrt{3}))$ |
| $A_b$ | $\text{Uniform}(A_{b,r}\,(1-5\%\sqrt{3}),\, A_{b,r}\,(1+5\%\sqrt{3}))$ |
| $V_{dep}$ | $\text{Uniform}(V_{dep,r}\,(1-5\%\sqrt{3}),\, V_{dep,r}\,(1+5\%\sqrt{3}))$ |
| $\psi$ | $\text{Uniform}(32°, 38°) \sim \text{Uniform}(35°\,(1-5\%\sqrt{3}),\, 35°\,(1+5\%\sqrt{3}))$ |

Finally, the uncertainty in $i_f$ and $A_b$ is hardly quantifiable given their computation method. For these reasons and homogeneity, the degree of uncertainty of $\psi$ and $V_{dep}$ is considered suitable also for $i_f$ and $A_b$.

The resulting uncertainty characterization is summarized in Table 1.

### 4.2  Uncertainty analysis of the event characteristics

The procedure used to assess the propagation of the uncertainty in $i_f$, $A_b$, $V_{dep}$ and $\psi$ on the event characteristics (i.e. $D$, $I$, $c$

and $E$) of each debris flow is schematized in Fig. 3(b) and it is composed of two main steps.

Fist, the input samples, namely the ordered sets of variable values in the form $(i_f, A_b, V_{dep}, \psi)$, must be obtained. These samples are generated by using the Latin Hypercube Sampling (LHS) (Fig. 3(b.1)), introduced by McKay et al. (2000). This method produces $N$ samples starting with a division of each variable uncertainty range into $N$ disjoint intervals of equal probability. Then, one value is randomly selected within every interval, thus obtaining $N$ values for each variable. These

values are then arranged in the LHS matrix, composed of $N$ rows and $k$ columns, where $k$ is the number of the variables (four, in the specific case). In each column, the $N$ values relevant to a single variable are inserted in random order (Helton et al., 2006). Each row of this matrix gives one of the $N$ variable samples. According to Marino et al. (2008), to ensure accuracy, the sample size $N$ should be at least greater than $k$. In this study, $N$ is set to 100 and the $(100 \times 4)$ LHS matrix is generated for each event, based on the previously established pdfs.

Second, the event characteristics are obtained starting from each input sample, resulting in 100 $(I, D)$ couples (Fig. 3(b.2)), together with the related $c$ and $E$ values, for each event. Therefore, the overall total of $(I, D)$ couples obtained is $100 \cdot 84 = 8400$, where $84$ is the number of considered debris flows.

### 4.3  Uncertainty analysis of the threshold

The uncertainties propagation in the threshold estimate is quantified with a further MC procedure. In this case, a sample is

generated selecting randomly one of the possible 100 $(I, D)$ couples for each event, resulting from the previous MC simulation (Fig. 3(c.1)). Hence, one sample consists of $84$ $(I, D)$ couples. Following this procedure, 5000 samples are created and used to estimate as many thresholds (Fig. 3(c.2)), namely 5000 $(a, b)$ couples.

## 5 Results and discussion

### 5.1 Variability of the event characteristics

As described in Sect. 4.2, the outputs of the first MC simulation applied to the dataset are 100 possible event characteristics (i.e. $D$, $I$, $c$ and $E$) for each debris flow. The relative variability of all these outputs is quantified through the computation of the $CV$ of each event characteristic distribution. This allows providing a complete inspection and interpretation of the all outputs. The absolute variability is then quantified through the computation of the variability range given by the difference between the minimum and the maximum values of the variable and it is evaluated only for the $D$ and $I$ distributions. This analysis allows highlighting the variability of the $(I, D)$ couples in the $ID$ plane for each event. Finally, possible correlations between the absolute variability and some event features are assessed.

### 5.1.1 Relative variability

The $CV$, by definition, is a standardized measure of dispersion (Håkanson, 2000) and allows comparing the relative variability of the results independently of their measurement units (Abdi, 2010) and of their means. For this reason, the $CV$ is chosen as the statistical quantity for comparing the relative variability between both the same characteristic of different events and different characteristics of the same event.

The $CV$s of the distributions of $D$, $I$, $E$ and $c$ for each event are shown in Table 2. In the following the trends and the differences in the $CV$s are highlighted and justified on the basis of some event aspects:

– the $D$ distributions have the largest and most variable $CV$s with respect to all the other event characteristics: $CV_D$ vary between $0\%$ and $157.5\%$. The reason for this behaviour will be clarified further on;

– the distributions of $I$ are characterised by a lower spread with respect to the $D$ distributions, being all the $CV_I$ values within $0\%$ and $30.0\%$. The reason for this behaviour is connected to the fact that $I$, by definition, is an average and therefore the effects of the variables uncertainty are smoothed by the averaging. Also the reason why $CV_I$ can be zero will be explained further on;

– also the concentration distributions show a low spread. As $c$ is characterised by an upper bound (i.e. $0.9\,c_b$), the $CV_c$ is strictly related to the proximity of $c_r$ to this maximum value and consequently, according to Eq. (2), to the value of $i_{f,r}$. As shown in Fig. 4, until the $i_{f,r}$ is less than about $0.3$, the $CV_c$ tends to go up by increasing the $i_{f,r}$ since $c_r$ is sufficiently smaller than $0.9\,c_b$. Instead, if the $i_{f,r}$ is between about $0.3$ and $0.4$, the $CV_c$ tends to decrease by increasing the $i_{f,r}$ as $c$ reaches the maximum and it is equal to $0.9\,c_b$ for an increasing number of $i_f$ samples. Finally, the $CV_c$ becomes $0\%$ if the $i_{f,r}$ is greater than about $0.4$ as $c$ is always equal to $0.9\,c_b$ independently from the values of the $i_f$ samples. However, even in the worst conditions in terms of variability, $CV_c$ is small and reaches a maximum value of $14.3\%$;

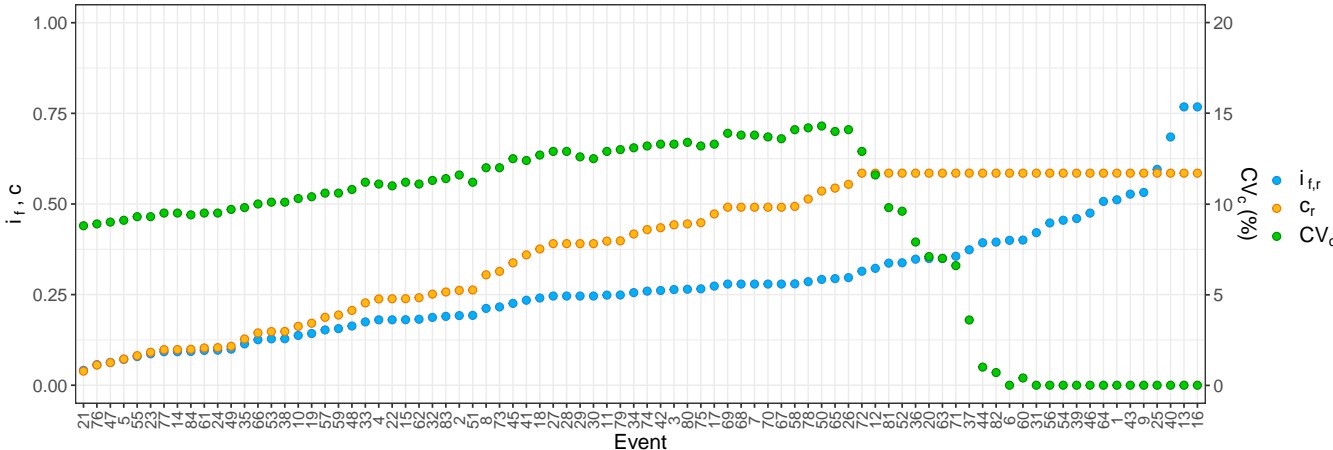

**Figure 4.** Comparison of the reference slope $i_{f,r}$, reference concentration $c_r$ and $CV_c$ trends. The events have been sorted with increasing slopes.

- the volume per unit area distributions shows $CV_E$s that vary between 7.1% and 64.6%. It is worth noting that high uncertainty in the estimation of $E$ does not necessarily imply large $CV_D$ and/or $CV_I$ (e.g. event 12) and vice versa (e.g. event 38). This suggests that the relative variability in $I$ and $D$ does not depend only on the relative variability in the needed rainfall volume per unit area but also on how the available rainfall volume is distributed into the hyetograph time intervals.

To better understand the variability in $D$ and $I$, we classify the events into three categories based on the $CV_D$ values:

1. events with zero variability: $CV_D = 0\%$;

2. events with low variability: $0\% < CV_D \leq 30\%$;

3. events with high variability: $CV_D > 30\%$.

The first category comprises 48 events for which the 100 simulations of the MC method have provided always the same $(I, D)$ couple. For these events, the propagation of the variables uncertainty does not affect the $(I, D)$ couple estimation resulting in $CV_D = CV_I = 0\%$. This type of result is related to two conditions:

- regardless of the variables uncertainty, the concentration is always equal to $0.9\,c_b$. In this case, also $CV_c$ is equal to zero (e.g. events 1, 13 and 25) and the variation in $E$ is only due to the propagation of $A_b$ and $V_{dep}$ uncertainty (Eq. (4)), namely $CV_E \simeq 7.1\%$. For these 14 events, such a small variation in $E$ results in the constant computation of the same $(I, D)$ couple;

- despite $CV_c$ is not zero and the $CV_E$ is greater than 7.1% (e.g. event 28), the condition of Eq. (8) is satisfied, in all the 100 simulations, considering always the same hyetograph time intervals. 34 events fall into this condition.

**Table 2.** Coefficients of variation of the event characteristics related to each debris flow expressed as a percentage. $CV_D$ is the coefficient of variation of the duration distribution, $CV_I$ of the intensity distribution, $CV_c$ of the concentration distribution and $CV_E$ of the rainfall volume per unit area distribution.

| Event | $CV_D$ | $CV_I$ | $CV_c$ | $CV_E$ | Event | $CV_D$ | $CV_I$ | $CV_c$ | $CV_E$ | Event | $CV_D$ | $CV_I$ | $CV_c$ | $CV_E$ |
|---|---|---|---|---|---|---|---|---|---|---|---|---|---|---|
| 1  | 0.0  | 0.0  | 0.0  | 7.1  | 29 | 0.0  | 0.0  | 12.6 | 26.6 | 57 | 0.0   | 0.0  | 10.6 | 15.6 |
| 2  | 34.6 | 5.5  | 11.6 | 18.3 | 30 | 0.0  | 0.0  | 12.5 | 26.9 | 58 | 157.5 | 14.9 | 14.1 | 35.4 |
| 3  | 34.2 | 7.9  | 13.3 | 30.7 | 31 | 0.0  | 0.0  | 0.0  | 7.1  | 59 | 0.0   | 0.0  | 10.6 | 15.9 |
| 4  | 23.0 | 6.2  | 11.1 | 17.8 | 32 | 0.0  | 0.0  | 11.3 | 18.4 | 60 | 0.0   | 0.0  | 0.4  | 8.8  |
| 5  | 7.1  | 1.5  | 9.1  | 12.6 | 33 | 13.8 | 4.1  | 11.2 | 17.2 | 61 | 16.1  | 4.4  | 9.5  | 13.4 |
| 6  | 0.0  | 0.0  | 0.0  | 7.1  | 34 | 30.9 | 9.7  | 13.1 | 28.1 | 62 | 0.0   | 0.0  | 11.1 | 18.1 |
| 7  | 39.6 | 13.5 | 13.8 | 36.3 | 35 | 24.8 | 22.7 | 9.8  | 14.2 | 63 | 0.0   | 0.0  | 7.0  | 55.5 |
| 8  | 26.4 | 2.8  | 12.0 | 20.9 | 36 | 0.0  | 0.0  | 7.9  | 58.4 | 64 | 7.8   | 1.0  | 0.0  | 7.1  |
| 9  | 0.0  | 0.0  | 0.0  | 7.1  | 37 | 0.0  | 0.0  | 3.6  | 34.2 | 65 | 0.0   | 0.0  | 14.0 | 43.5 |
| 10 | 26.6 | 15.7 | 10.3 | 14.7 | 38 | 74.7 | 30.0 | 10.1 | 14.4 | 66 | 0.0   | 0.0  | 10.0 | 14.4 |
| 11 | 0.0  | 0.0  | 12.9 | 27.0 | 39 | 0.0  | 0.0  | 0.0  | 7.1  | 67 | 110.1 | 28.2 | 13.6 | 35.5 |
| 12 | 0.0  | 0.0  | 11.6 | 55.9 | 40 | 0.0  | 0.0  | 0.0  | 7.1  | 68 | 32.9  | 4.6  | 13.8 | 35.9 |
| 13 | 0.0  | 0.0  | 0.0  | 7.1  | 41 | 35.3 | 15.1 | 12.4 | 24.7 | 69 | 37.1  | 12.3 | 13.9 | 36.2 |
| 14 | 15.6 | 2.6  | 9.5  | 13.2 | 42 | 0.0  | 0.0  | 13.3 | 30.0 | 70 | 0.0   | 0.0  | 13.7 | 36.1 |
| 15 | 0.0  | 0.0  | 11.2 | 17.8 | 43 | 0.0  | 0.0  | 0.0  | 7.1  | 71 | 0.0   | 0.0  | 6.6  | 54.8 |
| 16 | 0.0  | 0.0  | 0.0  | 7.1  | 44 | 0.0  | 0.0  | 1.0  | 15.0 | 72 | 0.0   | 0.0  | 12.9 | 54.7 |
| 17 | 0.0  | 0.0  | 13.3 | 34.6 | 45 | 25.3 | 10.0 | 12.5 | 22.8 | 73 | 16.0  | 0.6  | 12.0 | 21.6 |
| 18 | 0.0  | 0.0  | 12.7 | 25.0 | 46 | 0.0  | 0.0  | 0.0  | 7.1  | 74 | 0.0   | 0.0  | 13.2 | 29.5 |
| 19 | 13.3 | 3.0  | 10.4 | 15.1 | 47 | 16.7 | 4.9  | 9.0  | 12.1 | 75 | 0.0   | 0.0  | 13.2 | 31.4 |
| 20 | 41.7 | 4.7  | 7.1  | 54.9 | 48 | 0.0  | 0.0  | 10.8 | 16.4 | 76 | 37.7  | 18.9 | 8.9  | 12.3 |
| 21 | 0.0  | 0.0  | 8.8  | 11.7 | 49 | 15.9 | 4.3  | 9.7  | 13.1 | 77 | 31.1  | 9.4  | 9.5  | 13.1 |
| 22 | 0.0  | 0.0  | 11.0 | 17.8 | 50 | 39.9 | 11.0 | 14.3 | 42.4 | 78 | 0.0   | 0.0  | 14.2 | 38.7 |
| 23 | 0.0  | 0.0  | 9.3  | 13.0 | 51 | 25.1 | 8.6  | 11.2 | 18.9 | 79 | 0.0   | 0.0  | 13.0 | 26.4 |
| 24 | 17.3 | 5.7  | 9.5  | 13.3 | 52 | 0.0  | 0.0  | 9.6  | 63.1 | 80 | 25.2  | 9.4  | 13.4 | 31.1 |
| 25 | 0.0  | 0.0  | 0.0  | 7.1  | 53 | 35.4 | 1.6  | 10.1 | 14.5 | 81 | 0.0   | 0.0  | 9.8  | 64.6 |
| 26 | 62.4 | 16.9 | 14.1 | 44.8 | 54 | 0.0  | 0.0  | 0.0  | 7.2  | 82 | 0.0   | 0.0  | 0.7  | 10.1 |
| 27 | 72.1 | 25.7 | 12.9 | 26.3 | 55 | 22.6 | 10.6 | 9.3  | 12.7 | 83 | 0.0   | 0.0  | 11.4 | 18.4 |
| 28 | 0.0  | 0.0  | 12.9 | 26.2 | 56 | 0.0  | 0.0  | 0.0  | 7.1  | 84 | 16.1  | 1.0  | 9.4  | 13.2 |

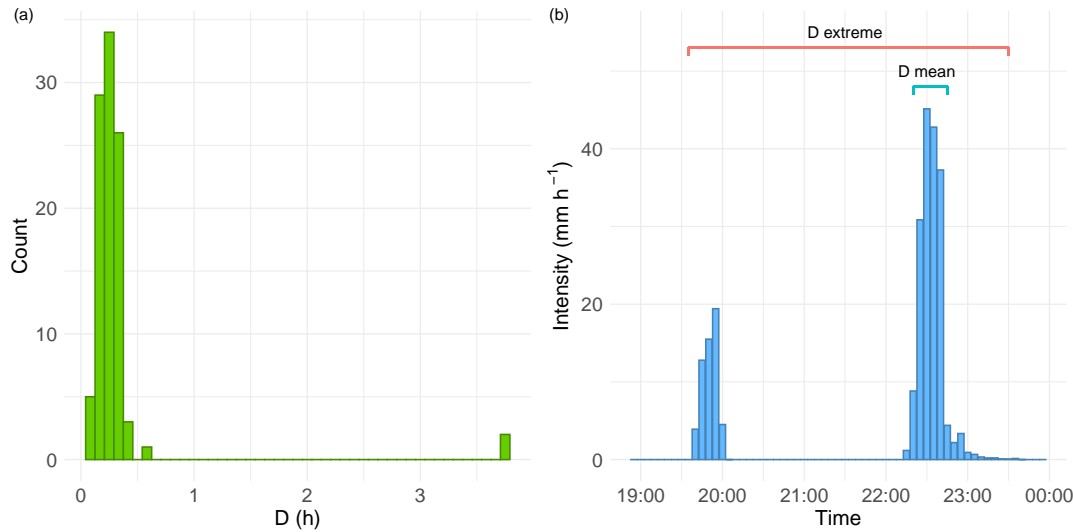

**Figure 5.** (a) Duration distribution histogram and (b) hyetograph of the event 58. The histogram shows the presence of an extreme isolated from the mass of the distribution. This extreme ($D = 3.75$ h) is shown in the hyetograph and compared with the mean ($D = 0.32$ h).

For the 19 events that belong to the second category, the variables uncertainty results in the computation of more $(I, D)$ couples that however are relatively close to the mean values: the $I$ and $D$ distributions are characterised by a standard deviation much smaller than the mean.

The third category includes 17 events for which the variables uncertainty implies high values of $CV_D$. This means that, for these events, the number of time intervals needed to satisfy the condition of Eq. (8) vary greatly with respect to the mean number: the variables uncertainty has a relative great impact on the computation of $D$. Moreover, the highest values of $CV_D$ highlight the presence of extremes in the $D$ distribution, namely of values of $D$ very distant from the mean. Indeed, $CV$ is very sensitive to the extremes (e.g. Chau et al., 2005; Arachchige et al., 2020), mainly if they are located in the right-hand tail of the distribution (Bendel et al., 1989). For instance, the effect of the extremes on the $CV_D$ is evident in event 58: the $D$ distribution of this event has an extreme much greater than the mean (Fig. 5(a)). This is due to the presence of zero intensity temporal instants in the middle of the hyetograph that must be considered (for two out of a hundred samples) to reach the highest values of $E$ (Fig. 5(b)). This results in a high value of the standard deviation with respect to the mean, namely a high value of the $CV_D$. It is worth noting that the effects of this condition on $I$ are smaller thanks to the mean carried out to obtain this event characteristic.

### 5.1.2 Absolute variability

The absolute variability ranges in the $ID$ plane of the $(I, D)$ couples for each event allows us to get an idea of how variable an event as a whole is and to presume how this variability may affect the threshold estimate. Consistently with the relative variability, the events with $CV_D = CV_I = 0\%$ have also zero-length absolute variability ranges. The non-zero ranges are shown

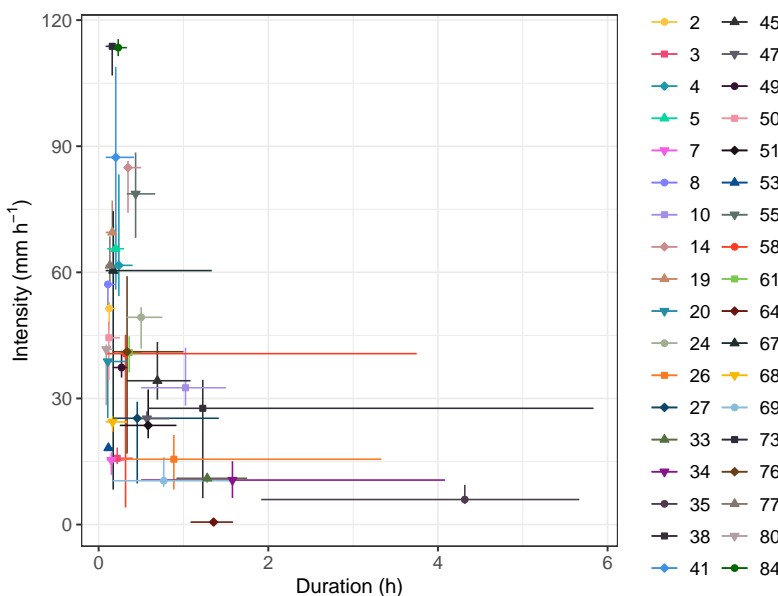

**Figure 6.** Plot of the absolute variability in the $(I, D)$ couples. The symbols are the mean values and the horizontal and vertical lines are respectively the duration and intensity variability ranges. To make the graph clearer, the events with uncertainty intervals equal to zero have not been represented and the linear scale is used for both axis.

in Fig. 6. As evident, the length of the ranges varies greatly depending on the event. In term of intensity, the maximum length is
66.21 mm h$^{-1}$ and it is reached with the event $67$ while, for the duration, the event $38$ is characterised by the maximum length that is equal to $5.25$ h. Besides, the length of the range for $D$ is less than 1 h in all but $8$ events while for $I$ it is less than 20 mm h$^{-1}$ in all but 7 events. Moreover, in most cases, the mean is located neither vertically nor horizontally in the middle of the variability ranges, namely the $D$ and $I$ distributions are asymmetrical. To quantify the asymmetry of the $I$ and $D$ distributions, the related skewness $SK_D$ and $SK_I$ are computed for each event and shown in Fig. 7. The events with zero variability are
characterised by $SK_D = 0$ and $SK_I = 0$. Moreover, in most cases, $SK_D$ are positive while $SK_I$ are negative: the longest tail of the distributions of $D$ and $I$ tends to be located on the right and the left of the mean respectively. This suggests that, given an event, the majority of the $D$ values are characterized by duration shorter than the mean and the greatest contribution to the absolute variability is given by the longest durations (i.e. by the $D$ distribution right extremes) as in event 38. Consistently, comparing Fig. 7 and Table 2, the events with the highest positive $SK_D$ are the events with the highest $CV_D$ (e.g. events $58$
and 67). Instead, given an event, the concentration of the intensity values is greater towards the highest values and the smallest intensities (i.e. the $I$ distribution left extremes) mostly contribute to the absolute variability (e.g. event 20 and 73). However, as said before, the $I$ extremes have a slight impact on the absolute variability of $I$ thanks to the mean procedure necessary for its computation that reduces the interval ranges.

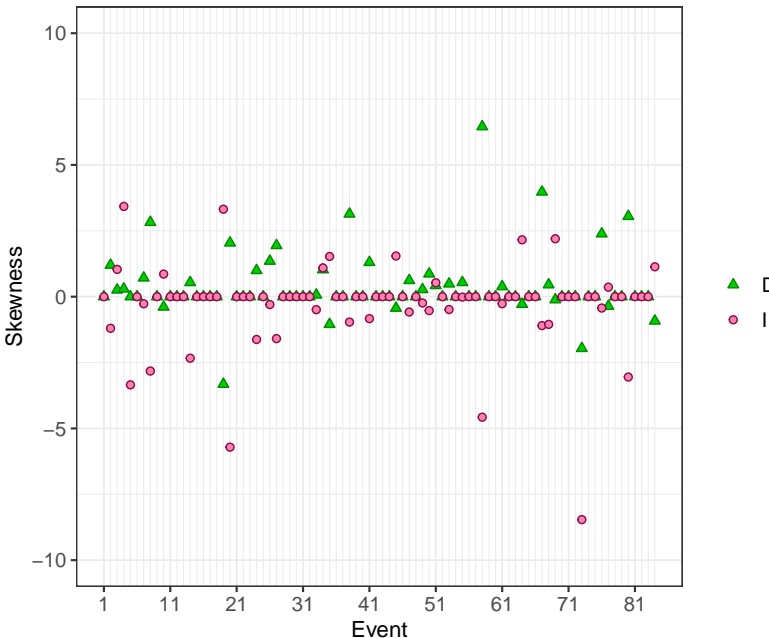

**Figure 7.** Skewness of the distributions of $D$ and $I$ for each event.

### 5.1.3 Correlation between the absolute variability and some event features

Despite the specificity of each considered event, it's possible to identify some event features that are correlated with the $D$ and $I$ absolute variability. It is worth noting that, in general, correlation does not imply causation (Wiedermann and Von Eye, 2016) but it is a starting point to understand if causality between the variables can be established.

We define $E_{av}$ as the rainfall volume per unit area available in the "main part of the hyetograph", namely the integral of the rainfall intensity on the smallest time interval comprising the peak and included between two instants with null intensities. We

can then introduce the ratio $E_r/E_{av}$. As shown in Fig. 8, the absolute variability of $D$ and $E_r/E_{av}$ are positive correlated. A small value of $E_r/E_{av}$ means that the main part of the hyetograph is amply able to provide $E_r$ (i.e. to satisfy the condition of Eq. (8) in the reference conditions). This tends to avoid having to consider null intervals to achieve the values of $E$ resulting from the MC simulation, namely to avoid $D$ extremes. The opposite situation occurs if the ratio takes high values.

Regarding the intensity, we define $I_{max}$ as the hyetograph maximum intensity and $I_{mean}$ as the the mean intensity of the

295 main part of the hyetograph for each event. The ratio $I_{max}/I_{mean}$ provides a quantitative measure of the shape of the event hyetograph or, equivalently, of how much impulsive the event is. As shown in Fig. 9, a positive correlation subsists between the non-zero absolute variability of $I$ and $I_{max}/I_{mean}$. If the shape of the hyetograph around the peak is flat, and the ratio $I_{max}/I_{mean}$ is low, the variability of $I$, connected to the variability of $D$, is small since the average procedure, necessary to compute $I$, involves similar intensities intervals. The opposite occurs when the event is impulsive and the ratio is high. This

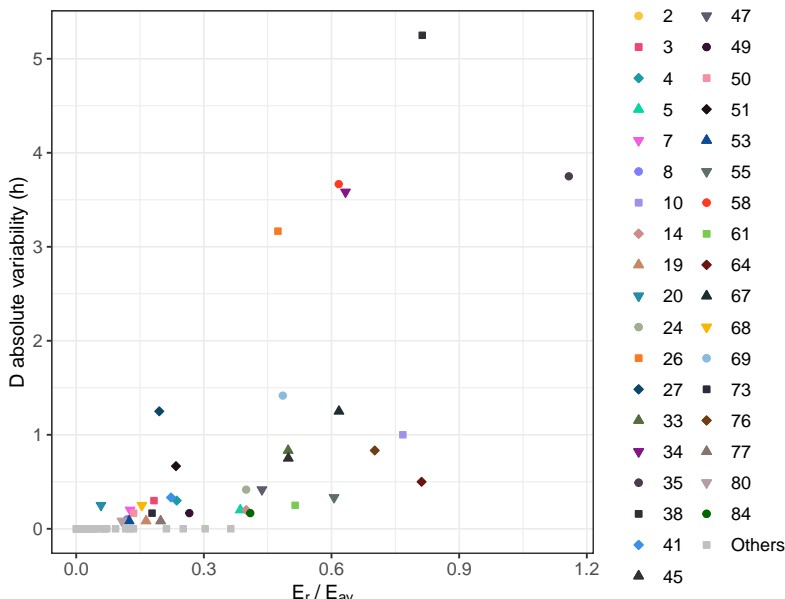

**Figure 8.** Positive correlation between the absolute variability of $D$ and $E_r/E_{av}$, where $E_{av}$ is the rainfall volume per unit area available in the main part of the hyetograph. Spearman correlation coefficient equals to $0.82$ $(p < 2.2 \times 10^{-16})$. To make the graph clearer, the events with absolute variability of $D$ equals to zero are represented with the same symbol.

**Table 3.** Mean, standard deviation, variation coefficient $CV$ and mean $95\%$ confidence interval $CI$ of the coefficients $a$ and $b$ of Eq. (12), computed performing the second MC simulation.

| Coefficient | Mean | Standard deviation | CV (%) | 95% mean confidence interval |
|:-----------:|:----:|:------------------:|:------:|:----------------------------:|
| $a$ | 6.0056 | 0.3882 | 6.46 | 0.0108 |
| $b$ | 0.6834 | 0.0199 | 2.91 | 0.0006 |

consideration is valid only for events with non-zero absolute variability in $I$ and $D$ and tends to explain why some events with high variability in $D$ have small variability in $I$ (e.g. event 26).

## 5.2  Variability of the threshold

The result of the second MC simulation is $5000$ $(a,b)$ couples. The main statistical quantities of their distributions are given in Table 3. The relative variability is quantified through the $CV$ that is equal to $6.46\%$ for $a$ and $2.91\%$ for $b$. The low spread

nature of $a$ and $b$, highlighted by the small $CV$ values, is also evident in the scatter plot and in the $3D$ histogram, respectively shown in Fig. 10(a) and 10(b). In addition, to analyse the absolute variability of the $I$-$D$ threshold relation, the intensity values for each $(a,b)$ couple are calculated for $D$ values spanning from five minutes to six hours with a five minutes time step. In this way, for each duration, we obtain an intensity distribution composed of 5000 samples. Then, the 2.5 and 97.5 percentiles

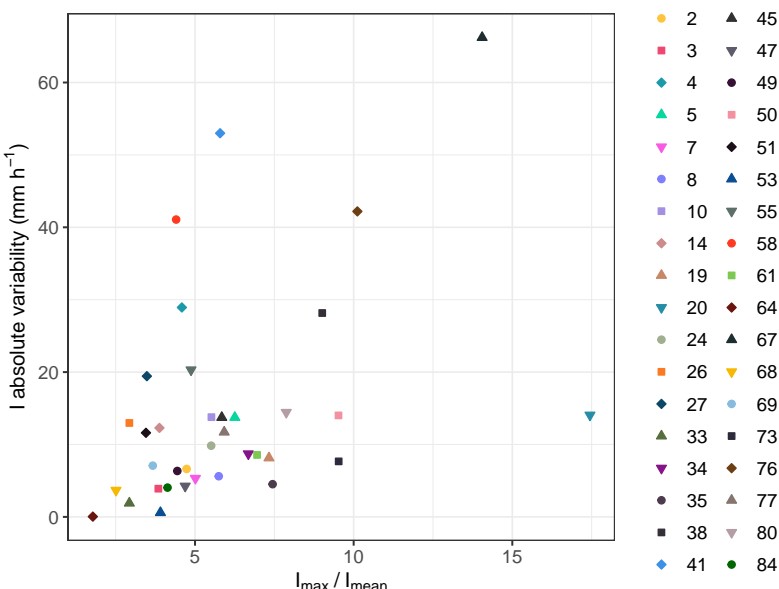

**Figure 9.** Positive correlation between the non-null absolute variability of $I$ and $I_{max}/I_{mean}$. $I_{max}$ is the maximum intensity and $I_{mean}$ is the mean intensity of the main part of the hyetograph. Spearman correlation coefficient equals to $0.48$ ($p = 0.0035$).

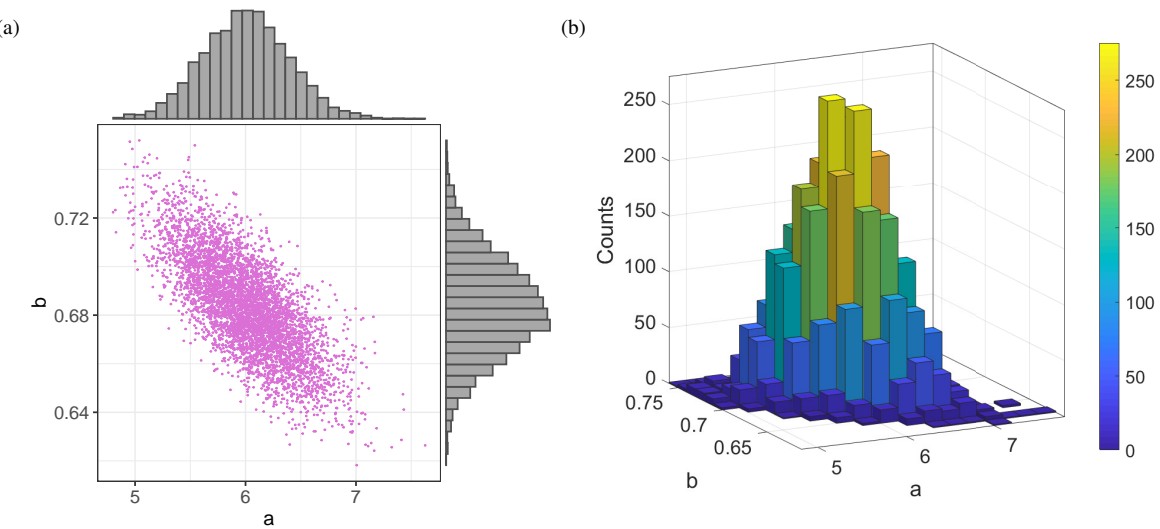

**Figure 10.** Values $a$ and $b$ of Eq. (12) obtained performing 5000 MC simulations: (a) scatter-plot and histogram and (b) $3D$ histogram

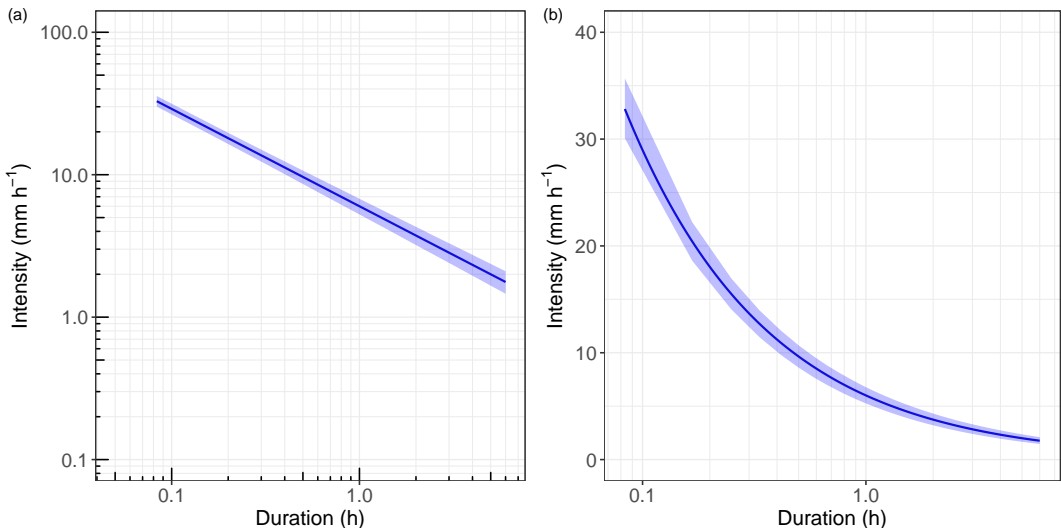

**Figure 11.** (a) Log-log and (b) semi-log plot of the threshold absolute variability. The blue line is the rainfall threshold obtained using the mean value of $a$ and $b$ (Table 3). The shaded area represents the threshold absolute variability whose upper and lower bounds have been computed considering the $2.5$ and $97.5$ percentiles of the intensity distributions for fixed durations.

of these distributions are chosen as upper and lower bounds of the threshold absolute variability. The result is shown in Fig. 11. According to the substantially symmetrical distributions of $a$ and $b$ (Fig. 10(a)), the threshold computed with the mean values of $a$ and $b$ (Table 3) is essentially equidistant from the lower and upper bounds. The variability bandwidth decreases monotonically by increasing the duration and varies between $5.61$ mm h$^{-1}$ and $0.64$ mm h$^{-1}$.

Hence, both the relative and the absolute variability highlight that the effect of the uncertainty in the variables on the threshold estimate is small. This is mainly due to the zero variability in the $D$ and $I$ distributions of $48$ events out of $84$: since the $(I, D)$ points of these events are located in the same positions in all the $5000$ MC simulations, they propagate zero uncertainty in the threshold computation.

## 5.3 Reference values versus MC means

Finally, a comparison between the results of the first and second MC simulation and the reference values is carried out. In particular, we compare:

- the means of the $D$ and $I$ distributions (Fig. 6), obtained with the first MC simulation, to the corresponding reference ones, for each event;

- the mean threshold (i.e. threshold computed with the mean values of $a$ and $b$) and the threshold absolute variability bounds (Fig. 11), resulting from the second MC simulation, to the reference threshold (i.e. Eq. (13)).

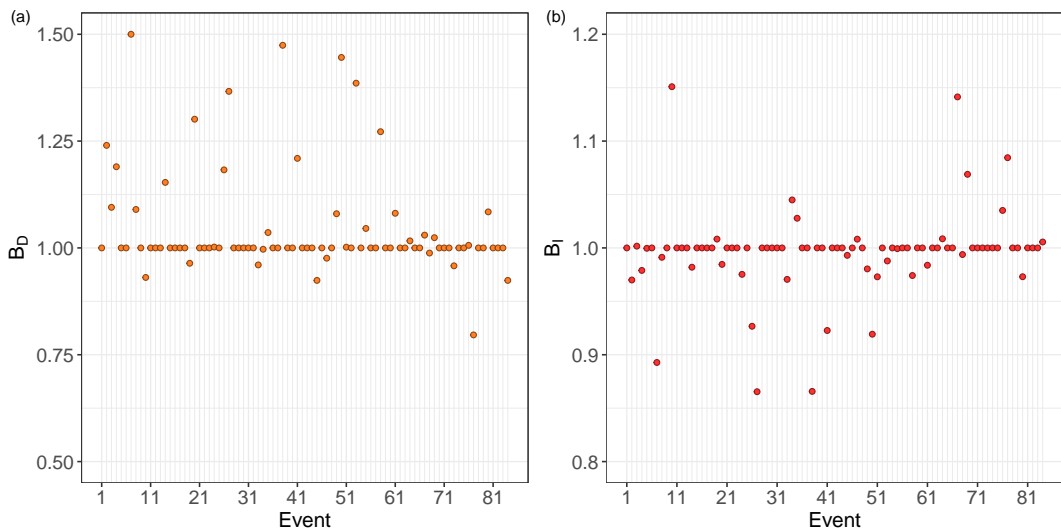

**Figure 12.** Bias of (a) duration $B_D$ and (b) intensity $B_I$ between the mean values of the $D$ and $I$ distributions obtained performing the first MC simulation and the corresponding reference values for each event.

As regards $D$ and $I$, according to Marra (2019), the bias of duration $B_D$ and intensity $B_I$ are computed for each event as:

$$B_D = \frac{D_m}{D_r} \qquad B_I = \frac{I_m}{I_r} \tag{14}$$

where the subscripts $m$ represent the mean of the MC $D$ and $I$ distributions. The result is shown in Fig. 12: $B_D$ deviates between $0.8$ and $1.5$ (Fig. 12(a)) while $B_I$ between $0.86$ and $1.15$ (Fig. 12(b)). Consistently with the variability analysis described in Sect. 5.1, most events (48) are characterised by $B_D = B_I = 1$. This means that for these zero-variability events, the reference duration and intensity are exactly the MC mean values of $I$ and $D$, namely the only MC $(I, D)$ couple. Moreover, most of the remaining events have $B_D > 1$ and $B_I < 1$. This signifies that the MC $(I, D)$ mean couples tend to be located lower and more to the right than the reference ones in the log-log $ID$ plane.

Regarding the threshold, the differences between the MC intensities $I_{MC,k}$, where $k$ stands for mean, upper bound and lower bound, and the reference threshold ones $I_{t,r}$ are carried out for the same durations used to define the absolute variability of the threshold:

$$\text{Diff}(k, D) = I_{MC,k}(D) - I_{t,r}(D), \quad k = \text{mean, upper bound, lower bound} \tag{15}$$

The result is shown in Fig. 13(a). For almost all durations, the intensities of the mean threshold are slightly lower than the reference threshold ones: a positive difference occurs only for the first time interval. Consistently with the obtained $a$ and $b$ mean values (Table 3) and the $B_D$ and $B_I$ trends, in the log-log $ID$ plane the mean threshold is respectively slightly more downward translated and clockwise rotated than the reference one. Instead, the upper and lower bounds are respectively always higher and lower than the reference threshold.

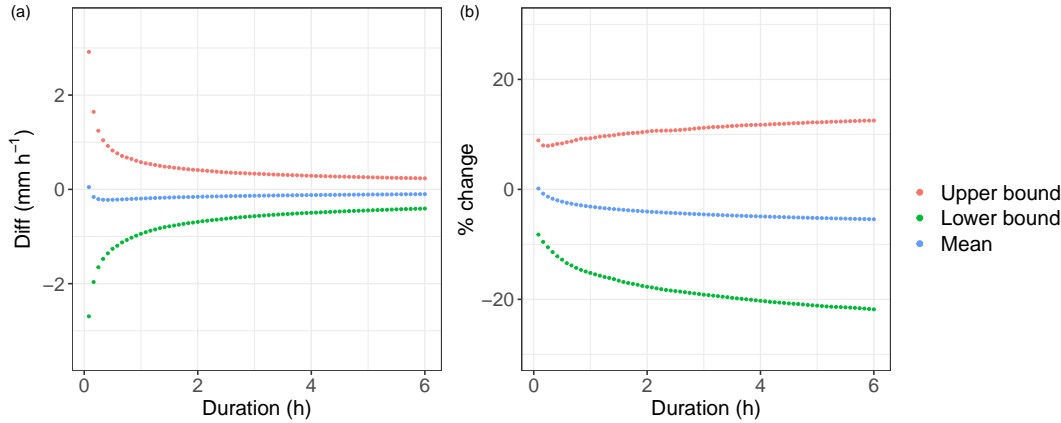

**Figure 13.** (a) Difference between the MC intensities (upper bound, lower bound and mean) and the reference threshold ones as a function of the duration; (b) percentage change of the MC intensities respect the reference threshold ones, as a function of the duration.

Subsequently, the percentage changes, defined as:

$$\% \text{ change}(k, D) = \left( \frac{\text{Diff}(k, D)}{I_{t,r}(D)} \right) \cdot 100, \quad k = \text{mean, upper bound, lower bound} \tag{16}$$

are computed to figure out how much the second MC outcomes deviate relatively from the reference threshold. The percentage changes are plotted in Figure 13(b): the mean threshold deviates between $0.14\%$ and $-5.44\%$, the upper bound between $8.06\%$ and $12.31\%$ and the lower bound between $-8.34\%$ and $-22.94\%$ from the reference one.

It can therefore be generally stated that the outcomes of the uncertainty analyses, both $(I, D)$ couples and threshold estimate, are consistent with the reference ones. Coherently with the previous analysis, also in this comparison, the duration is the quantity with the highest bias values. However, the mean threshold and the reference one are very close, pointing out the small effects of the differences between $D_m$ and $D_r$ on the threshold computation.

## 5.4 Further elements of uncertainty

In the calibration of the BDA-based threshold and in the assumptions of the developed method used to assess the uncertainty, it is possible to identify some elements that may introduce further uncertainty, beyond that considered in this analysis, in the calculation of the event characteristics and, consequently, in the estimate of the threshold. Firstly, the variability ranges and the probability distributions of the parameters and data, namely the uncertainty characterisation of the variables, are uncertain. Secondly, the equations of the BDA may be uncertain since they are based on some simplifications and hypothesis. Finally, the radar data may be affected by uncertainty due to other sources of error, beyond the beam shielding one (considered in this analysis), such as signal attenuation in heavy rain or wet radome attenuation (Marra et al., 2014). Nevertheless, at the present state of the research, it is not possible to assess the impact of these uncertainties on the event characteristics estimate and further study is required.

## 6 Conclusions

This study has aimed to assess the effects of the uncertainty in the physical and morphological parameters and data on the BDA-based threshold calibration to evaluate the method robustness. To that end, a suitable methodology composed of two MC cascade simulations has been developed and applied to a specific study area and dataset. The first MC simulation has allowed examining the uncertainty propagation in the event characteristics estimate. The results have highlighted that the $(I, D)$ couples can be affected by both high and low variability. Overall, the duration has found to be the most variable outcome in relative term while $I$, thanks to the average procedure, has a lower relative variability. In absolute term, the variability of the $(I, D)$ couples differs greatly between the events and the $D$ and $I$ distributions tend to be skewed to the right and left respectively. Moreover, considering the mean values of the events with non-zero variability (36 events out of 84), the uncertainty in the variables tends to provide slightly longer durations and slightly smaller intensities with respect to the reference ones. Notwithstanding, the second MC simulation has shown that the threshold computation is affected by small variability. The low dispersion of the threshold coefficients is mainly due to the 48 events with zero variability. As a result, the BDA method, applied to the considered dataset, can be described as robust since it provides a calibrated threshold low sensitive to the considered uncertainty in the parameters and data. This is also highlighted from the consistency between the uncertainty analysis mean threshold and the reference one.

Overall, the results of this analysis can be useful to calibrate a BDA-based threshold for a different study area since the investigation has highlighted the main critical elements that could undermine the BDA robustness. In particular, given a debris flow and the related rainfall event, it was noted that some event features are correlated with the variability of $D$ and $I$. The percentage of needed rainfall volume and available one in the main part of the hyetograph is positive correlated with the absolute variability of $D$. Moreover, the shape of the main part of the hyetograph, described by the ratio between the maximum and the mean intensity, is positive correlated with the non-null absolute variability of $I$. Therefore, given an event, these trends can be used to presume the possible variability in the estimate of $D$ and $I$, without carrying out a specific uncertainty analysis. In other words, if an event is characterised by (i) low availability of rainfall volume in the main part of the hyetograph with respect to the needed one and (ii) a peak intensity much greater than the mean one, variations in the the parameters and data is likely to result in high variability in $D$ and $I$ estimate. The presence of many events of this type could undermine the BDA robustness. Therefore, in these cases, it is advisable to put care in the estimate of the parameters and data.

Besides, given an event, further elements likely affecting the estimate of event characteristics have been highlighted in this study: (i) the variability ranges and probability distributions of the parameters and data, (ii) the equations constituting the BDA model and (iii) radar data. These elements can be affected by uncertainty and impact the event characteristics estimate. The uncertainty analysis performed in this study does not provide quantitative information on these impacts. Further analysis will assess how these three elements affect the $(I, D)$ couple estimate and, consequently, the threshold calibration.

Moreover, the developed method, composed of two cascade MC simulations, can be applied to assess the uncertainty related to other threshold calibration approaches whose event characteristics estimate is based not only on the hyetograph but also on other variables (e.g. the one proposed by Zhang et al. (2020)). Indeed, the developed method allows considering the entire range of uncertainty of the variables and, therefore, avoiding the analysis by scenarios, quite widespread in the literature for the

uncertainty analysis of rainfall thresholds (e.g. Nikolopoulos et al., 2014; Peres et al., 2018). Analysing by scenarios may not be

suitable if the uncertain parameters have a continuous range of variability. Indeed, a low number of input values combinations may not provide an overall assessment of the variability of the outputs.

Finally, it is worth noting that the results of this analysis are not useful to check the forecast capability of the threshold. Indeed, the variability in the threshold estimate due to the uncertainty of the inputs is not related to its forecast effectiveness but only to its robustness. The threshold forecast capability can be proved only by performing a proper validation analysis,

essential to make this tool operational. Since the calibration method applied to the specific study area is proved to be robust, further analysis will assess the forecast capability of the threshold, developing an appropriate validation method.

*Author contributions.* MM designed the experiments, performed the analysis and wrote the paper. DZ and GR supervised the study, analysed the results and wrote the paper.

*Competing interests.* The authors declare that they have no conflict of interest.

*Acknowledgements.* This work has been carried out within the project "Progetto WEEZARD: un sistema integrato di modellazione matematica a servizio della sicurezza nei confronti di pericoli idrogeologici in ambiente montano" (CARITRO Foundation – Cassa di Risparmio di Trento e Rovereto). We thank Prof. Siboni for providing valuable suggestions and Ripartizione Opere Idrauliche and Ufficio Idrografico, Provincia Autonoma di Bolzano (Italy) and Servizio Bacini Montani and Ufficio Previsioni e Pianificazione, Provincia Autonoma di Trento (Italy) for supplying radar and debris-flow data.

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
