# Peer review of "Uncertainty analysis of a rainfall threshold estimate for stony debris flow based on the Backward Dynamical Approach"

_Natural Hazards and Earth System Sciences, 2020_

## Referee Comment (RC1) · Anonymous Referee #1 · 10 Oct 2020

Summary The study descripts the uncertainty analysis of the estimation of rainfall threshold to initiate debris flow using the Backward Dynamical Approach. In general, the topic is novel and interesting to readers; whereas, quality of the manuscript is not up to the standards of an international journal. A major revision is needed to adjust the writing structure and clear the issues of the manuscript.

Major issues 1. The title used "stony" debris flow; it is better to explain why the study emphasize on the type of debris flow solely. 2. The Discussion content is combined with the Conclusion and poor discussed. Authors should make well discussions of your study and adjust your sections. It is suggested to adjust your writing structure in the

form of Results and Discussion, then Conclusion. The Conclusion contents need to be rewritten. 3. Authors should read the instructions of the journal to following its style. There are many parts in the manuscript is not coincided with the journal. 4. In P2, there are many method contents here; it is better to combine with the following Method section. The final paragraph in this page is not required and can be deleted. 5. In the manuscript, the uncertainty of rainfall threshold is estimated; but, can the authors tell us how the results improve our precision of rainfall threshold for issuing debris flow warning?

More minor issues 1. The term debris-flow and debris flow are mixed used in the context. 2. Figure 2 is not required and can be deleted. 3. In P4, what are t1 and t2 should be explained here briefly. 4. The texts (a), (b) etc. and legend are better to move on the corner of the figures 4, 9, and 10 to spare the space.

Opinion In general, the manuscript is interesting to readers of the journal. Whereas, the writing structure of the manuscript needs to be adjusted and the scientific quality needs to be improved. I prefer not to accept the manuscript at present form without a major revision.
* * *

---

## Referee Comment (RC2) · Anonymous Referee #2 · 9 Nov 2020

Dear Authors, Dear Editor,

I have read and carefully evaluated the manuscript "Uncertainty analysis of the estimation of stony debris flow rainfall threshold: the application to the Backward Dynamical Approach" submitted for possible publication in NHESS. The manuscript applies a double Monte Carlo simulation to investigate the robustness of the recently proposed BDA model with respect to uncertainty of input factors required to derived rainfall thresholds for stony debris flows. The idea is interesting and original and deserves attention. The English is generally clear. I identified a few shortcomings, as highlighted in my comments below. The manuscript could be considered again for publication after major

revisions.

—

GENERAL COMMENTS

1) The topic of rainfall threshold for landslides/debris flow initiation is intensely debated and many papers are published continuously. Most of the published papers have a low content of originality. I therefore suggest to better stress the elements of novelty in the proposed research. As instance:

- To my understanding, "stony debris flow" is a rather specific category. This could be briefly highlighted in the introduction and conclusion: most of the works abut rainfall thresholds mix different landslide typologies, others include DF and shallow landslides, others are addressed at DF in general (references could be easily found, e.g. with some review paper already in your reference list). You could highlight that studies explicitly addressed to stony DF are rare and thus more knowledge is needed on this field, hence suggesting the need of this test on the BDA model.

- You could expand the state of the art review and better link it to the originality (and usefulness) of your work. You cite a few relevant papers, but many other could be cited, especially in the central part of the introduction, to better set the stage for your work. And most of all, to avoid a sentence like "As stressed in the Introduction, the rainfall intensities i(t) associated with the event are assumed to be certain. Future analysis will assess and study also the uncertainties related to this piece of data", which seems an unnecessary justification. You could just say that uncertainty in rainfall threshold has been already investigated for temporal resolution (Marra 2019; Gariano et al., 2020, both already in your reference list), definition of the triggering rainfall (Peres et al. 2018), rain gauge selection (Abraham et al 2020) and so on. Whereas, a study is missing for the uncertainty in the parameters used by BDA for stony debris flows thresholds: nobody did it, this is (in my opinion) your main contribution to the progress of the state of the art. Abraham, M. T., Satyam, N., Rosi, A., Pradhan, B., & Segoni, S.

(2020). The Selection of Rain Gauges and Rainfall Parameters in Estimating Intensity-Duration Thresholds for Landslide Occurrence: Case Study from Wayanad (India). Water, 12(4), 1000. Peres, D. J., Cancelliere, A., Greco, R., & Bogaard, T. A. (2018). Influence of uncertain identification of triggering rainfall on the assessment of landslide early warning thresholds.

2) This paper is conceived and organized around some mathematical calculations. The risk is that the reader could perceive it as a "synthetic" experiment. I think it is important for NHESS readers to better put their minds on the specific case of study and it could be useful to add a brief description of the study case. This should include a brief description of the test area features and of the debris flows at hand. Also, some more information on the input data are needed (e.g. source of rainfall and debris flows datasets)

3) Discussions are almost missing, mainly they are mixed with the conclusions. I suggest providing separate sections. Discussions should contain an interpretation of the results, while in the conclusions you should summarize the lessons learnt. In particular, some points in my opinion are not clear enough: how does this study help us in predicting stony debris flows? Does it prove that BDA is robust, or does it prove that the utmost care should be put in calibrating/measuring the input parameters required? How does the uncertainty is reflected in the forecasting effectiveness of the resulting threshold? The latter point, in particular, is very important and some tests about that should be shown in the revised version of the manuscript.

SPECIFIC COMMENTS

L17. About review casualties, I suggest to add also the review "Dowling, C. A., & Santi, P. M. (2014). Debris flows and their toll on human life: a global analysis of debris-flow fatalities from 1950 to 2011. Natural hazards, 71(1), 203-227."

L23 I suggest "Usually, rainfall thresholds for debris flows initiation are power laws that link the rainfall duration to the rainfall cumulated or Intensity". This is because

when dealing with other landslide typologies, other parameters are often used (e.g. antecedent rainfall indexes).

L25 I would add also the work of Caine (1980), who started this methodology of analysis. Also, I would add some review. And I would substitute the work by Guzzetti by his work published the year after, reviewing ID thresholds (thus, more strictly related to your research).

L59 The Backward Dynamical Approach (BDA)...

L106: Since the study area is located in the Alps, this equation seems very low. Has this threshold been validated before (e.g. in Rosatti et al., 2019)? Could you report the validation result? Before going on with the reading, the readers should know how reliable this threshold is. How I said, this value seems very low to me and maybe another information should be provided about how the rainfall threshold is operated. E.g. within a long rainfall event a shorter but more intense burst of rain could easily reach the hourly peak intensity of 6.2 or higher. In that case is the threshold exceeded or not?

L112:"some": which one? The ones in tb. 1?

L157-160. This is not clear to me. If I understood correctly, in section 3.1 you set N=100, obtaining 100 random "points" in the I/D plane. Now, you randomly pick one of the I/D points, and you do it 5000 times. I guess most of the points are sampled many times. Because 5000 » 100. Moreover I do not understand how you can generate a threshold for each one of the selected points (5000 thresholds), since you should use many points to define a threshold (in short: many ID couples are needed to define a single a-b couple). I think I misunderstood something in this part, therefore I suggest to rephrase or to explain better.

L175. For what concerns = regarding?

L209: please check: the flow of the text is broken by the table and the image. It is hard

to follow (similar issues elsewhere)

L238-239: please check the text: possible issues.

GRAPHICAL IMPROVEMENTS

Figure 2: I strongly suggest adding another panel to this figure, where the DTM of the study area could be shown together with other relevant data (e.g. debris flows locations).

Please check the text immediately before and after images and tables. Sometimes sentences are split.

———————————————————

---

## Short Comment (SC1) · 17 Nov 2020

The paper by Martinengo et al. investigates the uncertainty in the determination of the debris-flow rainfall threshold based on the Back Dynamical Approach (BDA) proposed by Rosatti et al. (2019). The uncertainty analysis is performed through two Monte Carlo cascade simulations. The objectives of these simulations are (i) providing a sensitivity analysis of the BDA parameters and (ii) quantifying the impact of the variability of the rainfall estimate on the threshold parameters. Results highlight that the variability in the rainfall condition estimate is strongly related to the debris-flow characteristics and the to the hyetograph shape, while the threshold parameters are characterized by a

low statistical scattering. The paper diligently presents the uncertainties of the BDA method. However, it neglects some theoretical limitations of this method that are, in my opinion, important uncertainties to discuss. Here following the main points I would recommend addressing to provide a complete, physically-based (i.e., based on both analytical framework and experimental observations) uncertainty analysis.

1) More details are needed about the dataset (e.g., location of debris flows and rain gauges, data source for the volumes, etc.), possibly as a supplementary material. One minor observation related to this point: "stony debris flow" is rarely used in the debris-flow community. If the authors are referring to debris flows characterized by coarse material, especially at the front, and to non-cohesive mixtures, I would suggest clarifying this point in the dataset description stating that they are analyzing "collisional-frictional debris flows".

2) Direct observations collected in many Alpine basins (Bel et al., 2017; Coviello et al., 2020; Nikolopoulos et al., 2014) contradict the assumptions of uniform rainfall over the basin and of certain rainfall intensities associated with debris flows. Rainfall data gathered in the Gadria basin, that is located in the study area of the paper, can be used to discuss the uncertainties descending from these assumptions (Coviello et al., 2020). At Gadria, two different rain gauges located in the headwaters (Malga Gadria and Spartiacque) can be used for defining a critical rainfall threshold and data show a significant variability of both measured intensity and duration (see Fig. 1 from Marchi et al., 2019). For instance, the debris flow of 10 July 2017 shows very different I-D values at the two rain gauges while a pure rainfall analysis does not explain the initiation of the debris flow of 26 July 2016 that is characterized by a very low rainfall intensity recorded by only one rain gauge.

3) "Since being greater than zero is the only constraint of the other parameters, for homogeneity this CV value is considered suitable for all parameters". Field evidences and data show that a variation coefficient equal to 5% strongly underestimates the real variability of both basin area and deposited volume (Hürlimann et al., 2019). In
many monitored basin worldwide, it has been observed that the actual area contributing to debris flow initiation is considerably smaller than the whole basin area (Berti et al., 2020). Debris-flow volume estimations significantly differ - up to 30% - when performed through a digital elevation model of difference analysis, compared to the time-integration of the debris-flow discharge estimates (Coviello et al., 2020).

4) A constant sediment concentration of 0.65 in the bed of all debris-flow channels is questionable. Field studies show that sediment concentration and rheo-physical properties of debris flows feature a significant variability also during a single event (see Hürlimann et al., 2019 and reference therein).

Best regards, Velio Coviello

References

Bel, C., Liébault, F., Navratil, O., Eckert, N., Bellot, H., Fontaine, F. and Laigle, D.: Rainfall control of debris-flow triggering in the Réal Torrent, Southern French Prealps, Geomorphology, 291, 17–32, doi:10.1016/j.geomorph.2016.04.004, 2017.

Berti, M., Bernard, M., Simoni, A. and Gregoretti, C.: Physical interpretation of rainfall thresholds for runoff-generated debris flows, J. Geophys. Res. Earth Surf., 1–25, doi:10.1029/2019JF005513, 2020.

Coviello, V., Theule, J. I., Crema, S., Arattano, M., Comiti, F., Cavalli, M., Lucia, A., Macconi, P. and Marchi, L.: Combining Instrumental Monitoring and High-Resolution Topography for Estimating Sediment Yield in a Debris-Flow Catchment, Environ. Eng. Geosci., XXVI(4), in press, 2020.

Hürlimann, M., Coviello, V., Bel, C., Guo, X., Berti, M., Graf, C., Hübl, J., Miyata, S., Smith, J. B. and Yin, H. Y.: Debris-flow monitoring and warning: Review and examples, Earth-Science Rev., 199(December), 102981, doi:10.1016/j.earscirev.2019.102981, 2019.

Marchi, L., Coviello, V., Comiti, F., Crema, S., Cavalli, M. and Macconi, P.: Rainfall

threshold for debris flow occurrence in the Gadria catchment, eastern Italian Alps, Geophys. Res. Abstr., 21(July 2016), EGU2019-7188, 2019.

Nikolopoulos, E. I., Crema, S., Marchi, L., Marra, F, Guzzetti, F. and Borga, M.: Impact of uncertainty in rainfall estimation on the identification of rainfall thresholds for debris flow occurrence, Geomorphology, 221, 286–297, doi:10.1016/j.geomorph.2014.06.015, 2014.

Rosatti, G., Zugliani, D., Pirulli, M., and Martinengo, M.: A new method for evaluating stony debris flow rainfall thresholds: the Backward Dynamical Approach, Heliyon, 5, doi:10.1016/j.heliyon.2019.e01994, 2019.

[Figure]

**Fig. 1.** Expert-based critical rainfall threshold at Gadria, Italy (Marchi et al., 2019).

---

## Author Comment (AC1) · 4 Dec 2020

*We thank Velio Coviello for the comments provided that will be taken into account to improve the manuscript.*

The paper by Martinengo et al. investigates the uncertainty in the determination of the debris-flow rainfall threshold based on the Back Dynamical Approach (BDA) proposed by Rosatti et al. (2019). The uncertainty analysis is performed through two Monte Carlo cascade simulations. The objectives of these simulations are (i) providing

a sensitivity analysis of the BDA parameters and (ii) quantifying the impact of the variability of the rainfall estimate on the threshold parameters. Results highlight that the variability in the rainfall condition estimate is strongly related to the debris flow characteristics and the to the hyetograph shape, while the threshold parameters are characterized by a low statistical scattering. The paper diligently presents the uncertainties of the BDA method. However, it neglects some theoretical limitations of this method that are, in my opinion, important uncertainties to discuss. Here following the main points I would recommend addressing to provide a complete, physically-based (i.e., based on both analytical framework and experimental observations) uncertainty analysis.

1. More details are needed about the dataset (e.g., location of debris flows and rain gauges, data source for the volumes, etc.), possibly as a supplementary material. One minor observation related to this point: "stony debris flow" is rarely used in the debris-flow community. If the authors are referring to debris flows characterized by coarse material, especially at the front, and to non-cohesive mixtures, I would suggest clarifying this point in the dataset description stating that they are analyzing "collisional-frictional debris flows".
   *We agree. More information on data used to perform the analysis will be inserted in the revised version of the manuscript. As correctly claimed by Velio Coviello, with the term "stony debris flow" we refer to a debris flow in which the presence of silt and/or clay in the mixture is negligible and the stress is dominated by the collision of the particles (e.g. Takahashi, 2009, Stancanelli et al., 2015, Bernard et al., 2019, Hürlimann et al., 2019). We will clarify this definition in the revised manuscript.*

2. Direct observations collected in many Alpine basins (Bel et al., 2017; Coviello et al., 2020; Nikolopoulos et al., 2014) contradict the assumptions of uniform rainfall over the basin and of certain rainfall intensities associated with debris

flows. Rainfall data gathered in the Gadria basin, that is located in the study area of the paper, can be used to discuss the uncertainties descending from these assumptions (Coviello et al., 2020). At Gadria, two different rain gauges located in the headwaters (Malga Gadria and Spartiacque) can be used for defining a critical rainfall threshold and data show a significant variability of both measured intensity and duration (see figure below from Marchi et al., 2019). For instance, the debris flow of 10 July 2017 shows very different I-D values at the two rain gauges while a pure rainfall analysis does not explain the initiation of the debris flow of 26 July 2016 that is characterized by a very low rainfall intensity recorded by only one rain gauge.

*As described in Rosatti et al., 2019, the rainfall data used in the analysis are radar rainfall data. Hence, the spatial variability of the rainfall over the basins has been considered and an overall hyetograph for each event has been computed. In the revised manuscript, we will clarify the data used in the analysis in a new specific section. However, as highlighted in the Introduction of the manuscript, the focus of this work is not on the uncertainty in rainfall estimation but the effect of some BDA inputs uncertainty (i.e. the average slope $i_f$, the basin area $A_b$, the deposited volume $V_{dep}$ and the dynamical friction angle $\psi$) on the threshold calibration. We are aware of the importance of the rainfall uncertainty in the threshold calibration. Nevertheless, addressing this topic is not an objective of this work since a lot of literature already investigates this aspect (e.g. Peres et al. 2018, Marra 2019, Abraham et al 2020, Gariano et al., 2020).*

3. "Since being greater than zero is the only constraint of the other parameters, for homogeneity this CV value is considered suitable for all parameters". Field evidences and data show that a variation coefficient equal to 5% strongly underestimates the real variability of both basin area and deposited volume. In many monitored basin worldwide, it has been observed that the actual area contributing to debris flow initiation is considerably smaller than the whole basin area (Berti

et al., 2020; Hürlimann et al., 2019). Debris-flow volume estimations significantly differ - up to 30% - when performed through a digital elevation model of difference analysis, compared to the time-integration of the debris-flow discharge estimates (Coviello et al., 2020).

*We agree with Velio Coviello that the source of sediments area of a debris flow, i.e. the triggering zone, is not equivalent to the whole area of the basin affected by the event. The triggering area is only a portion of the basin and its size and location depend on the triggering condition of the debris flow (e.g. diffused erosion, erosions mainly along the main channel, failure of existing protection structures, landslides). However, the BDA method is not interested in where and how the debris flow is generated but in the amount of water that is needed to convey downstream and to deposit, in a given zone of the basin, the surveyed amount of sediments. This amount of water is linked to the hydrological discharge (i.e. to the rainfall) and since a considered debris flow has reached its deposition area, the whole basin upstream this area has to be considered as the source of the hydrological discharge, as described in Rosatti et. al., 2019.*

*As regards the deposited volume $V_{dep}$, the time-integration of the debris-flow discharge is the volume of the mixture $V_{mix}$ and not $V_{dep}$ (i.e. the saturated deposited volume surveyed after the event). Instead, the DEM of Difference (DoD) analysis provides an estimate of $V_{dep}$. $V_{mix}$ is greater than $V_{dep}$ since, at least, part of the water volume (and of the fine fraction of the sediments) flows away during the deposition (Coviello et al., 2019, Rosatti et al., 2019). For this reason, the volumes obtained performing the time-integration of the debris-flow discharge and the DoD differs (up to $30\%$) and their difference is consistent with the volume that flows away, as described in Coviello et al., 2019. This highlights that, in general, the large deviation between $V_{mix}$ and $V_{dep}$ is not primarily due to the uncertainty in the estimate of $V_{dep}$ but it is mainly a consequence of the different definition of these two volumes.*

*Anyway, since the deposited volumes of the analysed events have been provided*

*by regional agencies, that do not use an univocal survey methodology, we were unable to accurately define the degree of uncertainty of this input. For this reason, we have assumed a constant value of the variation coefficient $CV$ for each event, as also done for the other inputs.*

*For definition, the $CV$ is the ratio between the standard deviation $\sigma$ and the mean $\mu$ of the considered probability density function. Assuming an uniform distribution, to impose $CV = 5\%$ leads to set the lower and upper bounds of the input parameters and data distributions equal to $(1 - 5\%\sqrt{3})\mu \simeq (1 - 8.7\%)\mu$ and $(1 + 5\%\sqrt{3})\mu \simeq (1 + 8.7\%)\mu$ respectively (see table 1 of the manuscript in which $\mu$ is the reference value of the considered input parameter/data). The absolute range of variation between the lower and the upper bound is $\sim 17.4\%$. Lacking additional data on the uncertainty, we considered satisfactory this range of variability for all inputs.*

4. A constant sediment concentration of 0.65 in the bed of all debris-flow channels is questionable. Field studies show that sediment concentration and rheo-physical properties of debris flows feature a significant variability also during a single event (see Hürlimann et al., 2019 and reference therein).

*We are sorry but we didn't understand what was meant with this sentence. Perhaps there is a misunderstanding in the definition of the concentration of the sediment in the bed. In the BDA approach, the bed concentration $c_b$ is the concentration of the sediments in the layer below the debris flow and therefore it is a characteristic that does not depend on the flow. On the other hand, it is true that during a debris flow event, the mixture concentration $c$ changes in time and space, as described for example in Jacob and Hungr, 2005 and Takahashi, 2009. Instead, to the best of our knowledge, there are not experimental or field investigations asserting that also $c_b$ changes significantly during a single event. Not even the mentioned paper of Hürlimann deals with this topic.*

Velio Coviello

Free University of Bozen-Bolzano, Faculty of Science and Technology, Italy
velio.coviello@unibz.it

References

Bel, C., Liébault, F., Navratil, O., Eckert, N., Bellot, H., Fontaine, F. and Laigle, D.: Rainfall control of debris-flow triggering in the Réal Torrent, Southern French Prealps, Geomorphology, 291, 17–32, doi:10.1016/j.geomorph.2016.04.004, 2017.

Berti, M., Bernard, M., Simoni, A. and Gregoretti, C.: Physical interpretation of rainfall thresholds for runoff-generated debris flows, J. Geophys. Res. Earth Surf., 1–25, doi:10.1029/2019JF005513, 2020.

Coviello, V., Theule, J. I., Crema, S., Arattano, M., Comiti, F., Cavalli, M., Lucia, A., Macconi, P. and Marchi, L.: Combining Instrumental Monitoring and High-Resolution Topography for Estimating Sediment Yield in a Debris-Flow Catchment, Environ. Eng. Geosci., XXVI(4), in press, 2020.

Hürlimann, M., Coviello, V., Bel, C., Guo, X., Berti, M., Graf, C., Hübl, J., Miyata, S., Smith, J. B. and Yin, H. Y.: Debris-flow monitoring and warning: Review and examples, Earth-Science Rev., 199(December), 102981,
doi:10.1016/j.earscirev.2019.102981, 2019.

Marchi, L., Coviello, V., Comiti, F., Crema, S., Cavalli, M. and Macconi, P.: Rainfall threshold for debris flow occurrence in the Gadria catchment, eastern Italian Alps, Geophys. Res. Abstr., 21(July 2016), EGU2019-7188, 2019.

Nikolopoulos, E. I., Crema, S., Marchi, L., Marra, F., Guzzetti, F. and Borga, M.: Impact of uncertainty in rainfall estimation on the identification of rainfall thresholds for debris flow occurrence, Geomorphology, 221, 286–297,
doi:10.1016/j.geomorph.2014.06.015, 2014.

Rosatti, G., Zugliani, D., Pirulli, M., and Martinengo, M.: A new method for evaluating stony debris flow rainfall thresholds: the Backward Dynamical Approach, Heliyon, 5, doi:10.1016/j.heliyon.2019.e01994, 2019.

[Figure]

*Abraham, M. T., Satyam, N., Rosi, A., Pradhan, B., and Segoni, S.: The Selection of Rain Gauges and Rainfall Parameters in Estimating Intensity-Duration Thresholds for Landslide Occurrence: Case Study from Wayanad (India). Water, 12(4), 1000, 2020.*

*Bernard, M., Boreggio, M., Degetto, M., and Gregoretti, C.: Model-based approach for design and performance evaluation of works controlling stony debris flows with an application to a case study at Rovina di Cancia (Venetian Dolomites, Northeast Italy). Science of the total environment, 688, 1373-1388, 2019.*

*Coviello, V., Theule, J. I., Marchi, L., Comiti, F., Crema, S., Cavalli, M., Arattano, M., Lucia, A., and Macconi, P.: Deciphering sediment dynamics in a debris flow catchment: insights from instrumental monitoring and high-resolution topography. Association of Environmental and Engineering Geologists; special publication 28. Colorado School of Mines. Arthur Lakes Library, 2019.*

*Gariano, S. L., Melillo, M., Peruccacci, S., and Brunetti, M. T.: How much does the rainfall temporal resolution affect rainfall thresholds for landslide triggering?. Natural Hazards, 100(2), 655-670, 2020.*

*Jakob, M., and Hungr, O.: Debris-flow hazards and related phenomena (Vol. 739). Berlin: Springer, 2005. Marra, F.: Rainfall thresholds for landslide occurrence: systematic underestimation using coarse temporal resolution data, Natural Hazards, 95, 883–890, 2019. Peres, D. J., Cancelliere, A., Greco, R., and Bogaard, T. A.: Influence of uncertain identification of triggering rainfall on the assessment of landslide early warning thresholds, 2018.*

*Stancanelli, L. M., Lanzoni, S., and Foti, E.: Propagation and deposition of stony debris flows at channel confluences. Water Resources Research, 51(7), 5100-5116, 2015.*

*Takahashi, T.: A review of Japanese debris flow research. International Journal of Erosion Control Engineering, 2(1), 1-14, 2019.*

---

## Author Response (AR1)

**Relevant changes made in the manuscript**

In the revision of the manuscript, we have followed the suggestions provided by the reviewers addressing all the comments and requests. To this end, we have heavily modified the paper, clarifying some aspects and introducing new elements. In particular, according to the check list of the main issues highlighted by the Editor, the principal changes made in the manuscript are the following:

1. the Introduction has been rewritten to better context this work in the state of the art of the rainfall threshold uncertainty analysis and to clarify the aim of this study. In this way, the references have been expanded and the novelties of our work have been highlighted;

2. we have made the description of the BDA-based threshold calibration clearer, highlighting its assumptions and the consequent meaning of its data and parameters (Sect. 2);

3. we have included a specific section for the study area and data (Sect. 3) to describe the dataset used in this analysis and to explain how the data and parameters were estimated in the calibration of the threshold already performed by Rosatti et al. (2019);

4. the uncertainty characterisation of the parameters and data is described in a specific section in which we better justify the assumptions made (Sect 4.1);

5. the discussion has been separated from the conclusions. We have chosen to incorporate the discussion in the results to combine the outcomes presentation, analysis and interpretation (Sect. 5). Moreover, we have provided a more detailed interpretation of the results both expanding the already present considerations and introducing further analyses (i.e. skewness analysis and correlation between the absolute variability and some event features, Sect. 5.1.2 and 5.1.3);

6. the conclusions have been completely rewritten to highlight the key findings of the analysis as well as how the developed method and the obtained results can be useful for further studies;

7. in the Conclusions, we have highlighted that the uncertainty analysis does not provide any indication on the forecasting effectiveness of the threshold but it is a prerequisite for the validation analysis.

Here below are the comments of the referees #1 and #2 with the related already provided responses (in *italic blue font*) and the descriptions of the consequent changes actually made in the manuscript (in *italic brown font*).

**Referee #1**

*We would like to thank the anonymous referee #1 for the useful comments that will be used to improve the manuscript.*

**Summary**
The study descripts the uncertainty analysis of the estimation of rainfall threshold to initiate debris flow using the Backward Dynamical Approach. In general, the topic is novel and interesting to readers; whereas, quality of the manuscript is not up to the standards of an international journal. A major revision is needed to adjust the writing structure and clear the issues of the manuscript.

**Major issues**

1. The title used "stony" debris flow; it is better to explain why the study emphasize on the type of debris flow solely.

   *We agree. In our work, we only consider stony debris flow because of the hypothesis on which the BDA is based on. The BDA approach is applicable only if the presence of silt and/or clay in the mixture is negligible, namely if silt and/or clay have no effects on the overall rheological properties of the mixture. We will explain this aspect in the Introduction of the revised manuscript.*

   *In the Introduction of the revised manuscript, we have explained what stony debris flows are and why we refer only to this type of event (Lines $50 - 56$).*

2. The Discussion content is combined with the Conclusion and poor discussed. Authors should make well discussions of your study and adjust your sections. It is suggested to adjust your writing structure in the form of Results and Discussion, then Conclusion. The Conclusion contents need to be rewritten.

   *We will adjust the paper structure in the revised manuscript dividing discussion and conclusions and improving their contents as suggested. In particular, in the Discussion we will analyse and interpret the results of our study more in detail. We will better stress what are the main effects of the uncertainty of the input parameters on the rainfall conditions estimate (i.e. intensity I and duration D) and what are the event characteristics that emphasize the variability of this estimate. We will also better highlight the impacts of the input parameters uncertainty on the threshold estimate and the differences between the mean values of both the (I, D) couples and the threshold, obtained performing the Monte Carlo simulations, and the outcomes of the standard calibration (i.e. the reference values) and what these differences imply. The Conclusion will be rewritten focusing on the resulting robustness of the BDA method. We will also stress in which cases it is necessary to put care in the estimate of the input parameters, why it is advisable to apply the BDA to other study areas (namely to test this approach with different data), how it is possible to implement the developed uncertainty analysis to other parameters-dependent thresholds (e.g. the threshold proposed by Zhang et al., 2020) and finally, why it is im-*

*portant to perform a suitable validation analysis (not addressed in this manuscript) to assess the threshold forecast capability.*

*The discussion has been incorporated in the results section and a more detailed interpretation of the results have been provided. In particular, we have introduced (i) the skewness analysis (Sect. 5.1.2), (ii) the correlation between the absolute variability and some event features (Sect. 5.1.3) and (iii) some considerations on further elements of uncertainty (Sect. 5.4). The Conclusions have been rewritten, highlighting better the results of these analyses and the usefulness of the study, both to calibrate a BDA-based threshold for a different study area and to apply the developed method to assess the uncertainty in other threshold calibration approaches.*

3. Authors should read the instructions of the journal to following its style. There are many parts in the manuscript is not coincided with the journal. *We thank the referee for the indications. We will adjust the structure in the revised version of the manuscript following better the journal guideline.*

   *The structure of the manuscript has been adjusted introducing the study area and data section (Sect. 3) and incorporating the discussion in the results (Sect. 5).*

4. In P2, there are many method contents here; it is better to combine with the following Method section. The final paragraph in this page is not required and can be deleted. *We agree. We will reduce the description of the BDA and the proposed method for the uncertainty analysis in the Introduction. We will leave only a synthetic scheme of both because we consider it useful to guide the reader through the manuscript.*

   *The description of the method content has been reduced in the Introduction and expanded in Sect. 2 and 4.*

5. In the manuscript, the uncertainty of rainfall threshold is estimated; but, can the authors tell us how the results improve our precision of rainfall threshold for issuing debris flow warning? *Perhaps we were not clear enough in explaining the purpose of this paper. The uncertainty analysis and the validation differ in terms of purpose. The uncertainty analysis aims to inspect the robustness of a method, namely checking if the uncertainty of the inputs leads to high variability in the outputs. Instead, the validation purpose is to check the effectiveness of a method and the reliability of its outputs. This work proposes an operative methodology to analyse the robustness of the BDA method used for calibrating a threshold. Hence the results of this study cannot say anything about the reliability in forecasting debris flows. The goal of the BDA was to provide a physical-based approach to calibrate the rainfall threshold. However, the assessment of a possible improvement brought about by this new approach, with respect to other literature methods, must be carried out through a suitable validation analysis. We are currently developing a proper approach to validate the BDA-based threshold that will be the object of a forthcoming manuscript. In any case, we believe that it makes sense to validate only a threshold that revealed to be robust. Therefore, in carrying out our research, we first faced the robustness matter (this paper) and*

*then the will deal with the validation topic (forthcoming paper).*
*We will better clarify the aim of this work in the Introduction of the revised manuscript.*

*The purpose of this work has been better explained in the Introduction (Lines 59 − 65). Moreover, in the Conclusions, we have clarified that the uncertainty analysis cannot provide explicit indications on the forecast capabilities of the threshold but it is a necessary prerequisite for the validation (Lines 397 − 401).*

**Minor issues**

1. The term debris-flow and debris flow are mixed used in the context.
   *We thank the referee for the comment. Actually, we made some mistakes in the use of the two terms. In the revised manuscript, "debris-flow" and "debris flow" will be employed as an adjective and as a noun respectively.*

   *The use of the two terms has been revised in all the manuscript.*

2. 2. Figure 2 is not required and can be deleted
   *As suggested by the anonymous referee #2, in the revised version of the manuscript we will insert a section devoted to the study area and data. We think that Figure 2 should acquire more meaning in this new section.*

   *The Figure 2 has been moved in the study area and data section (Sect. 3).*

3. In P4, what are t1 and t2 should be explained here briefly.
   *We agree. We will insert a brief explanation of the computation of $t_1$ and $t_2$ in the revised version of the manuscript.*

   *In Sect. 2 we have explained more in detail what are $t_1$ and $t_2$ and how they are computed (Lines 94 − 108).*

4. The texts (a), (b) etc. and legend are better to move on the corner of the figures 4, 9, and 10 to spare the space.
   *We will follow the suggestion.*

   *The mentioned figures (Fig. 4, 10 and 13 in the revised manuscript) have been fixed.*

**Opinion**
In general, the manuscript is interesting to readers of the journal. Whereas, the writing structure of the manuscript needs to be adjusted and the scientific quality needs to be improved. I prefer not to accept the manuscript at present form without a major revision.

*Bibliography*

*Zhang, S. J., Xu, C. X., Wei, F. Q., Hu, K. H., Xu, H., Zhao, L. Q., and Zhang, G. P.: A physics-based model to derive rainfall intensity-duration threshold for debris flow. Geomorphology, 351, 106930, 2020.*

**Referee #2**

*We appreciate the useful comments provided by the anonymous referee #2 that will be used to improve the manuscript.*

Dear Authors, Dear Editor, I have read and carefully evaluated the manuscript "Uncertainty analysis of the estimation of stony debris flow rainfall threshold: the application to the Backward Dynamical Approach" submitted for possible publication in NHESS. The manuscript applies a double Monte Carlo simulation to investigate the robustness of the recently proposed BDA model with respect to uncertainty of input factors required to derived rainfall thresholds for stony debris flows. The idea is interesting and original and deserves attention. The English is generally clear. I identified a few shortcomings, as highlighted in my comments below. The manuscript could be considered again for publication after major revisions.

**General comments**

1. The topic of rainfall threshold for landslides/debris flow initiation is intensely debated and many papers are published continuously. Most of the published papers have a low content of originality. I therefore suggest to better stress the elements of novelty in the proposed research.
   As instance:

   - To my understanding, "stony debris flow" is a rather specific category. This could be briefly highlighted in the introduction and conclusion: most of the works abut rainfall thresholds mix different landslide typologies, others include DF and shallow landslides, others are addressed at DF in general (references could be easily found, e.g. with some review paper already in your reference list). You could highlight that studies explicitly addressed to stony DF are rare and thus more knowledge is needed on this field, hence suggesting the need of this test on the BDA model.

   - You could expand the state of the art review and better link it to the originality (and usefulness) of your work. You cite a few relevant papers, but many other could be cited, especially in the central part of the introduction, to better set the stage for your work. And most of all, to avoid a sentence like "As stressed in the Introduction, the rainfall intensities i(t) associated with the event are assumed to be certain. Future analysis will assess and study also the uncertainties related to this piece of data", which seems an unnecessary justification. You could just say that uncertainty in rainfall threshold has been already investigated for temporal resolution (Marra 2019; Gariano et al., 2020, both already in your reference list), definition of the triggering rainfall (Peres et al. 2018), rain gauge selection (Abraham et al 2020) and so on. Whereas, a study is missing for the uncertainty in the parameters used by BDA for stony debris flows thresholds: nobody did it, this is (in my opinion) your main contribution to the progress of the state of the art. Abraham, M. T., Satyam, N., Rosi, A., Pradhan, B., & Segoni, S. (2020). The Selection of Rain Gauges

and Rainfall Parameters in Estimating Intensity-Duration Thresholds for Landslide Occurrence: Case Study from Wayanad (India). Water, 12(4), 1000. Peres, D. J., Cancelliere, A., Greco, R., & Bogaard, T. A. (2018). Influence of uncertain identification of triggering rainfall on the assessment of landslide early warning thresholds.

*We agree with this comment. In the Introduction, we will highlight that we refer only to stony debris flow because of the assumptions on which the BDA is based on and we will also clarify what we mean by "stony debris flow", as suggested by the anonymous referee #1.*
*The novelty of our work will be better stressed in the revised version of the manuscript to make it clearer to the reader. As correctly asserted by the referee, in the literature there are other works that investigate the uncertainty in rainfall thresholds estimate. The uncertainty analysis of these works are mainly focused on rainfall since the classical approaches, used to calibrate the threshold, estimate the rainfall condition related to an event only on the basis of the hyetograph. Instead, the BDA method computes the rainfall condition related to an occurred event on the basis of a schematic description of the phenomenon dynamics. This means that the calibration of the threshold starts from the estimate of some physical and morphological parameters and surveyed data. To assess the robustness of the BDA requires a proper method that allows to quantify the effects of the input parameters and data uncertainties on the threshold estimate. If these effects are low, the BDA can be defined as robust. The main novelties of our work are therefore: a new method of assessing the robustness of a rainfall threshold (based on two cascade Monte Carlo simulations) and the application of this method to the BDA. It is worth noting that the developed approach can be also applied to other threshold calibrations that use physical-based parameters (e.g. the one proposed by Zhang et al., 2020).*
*In the Introduction, to better stress the novelties of our work, we will insert other literature references concerning the uncertainty analysis of rainfall thresholds and we will compare our work to the state of the art.*

*In the revised manuscript, the Introduction has been rewritten to better highlight (i) the state of art of the literature related to the uncertainty analysis of rainfall thresholds, (ii) why we refer only to stony debris flow and (iii) what are the novelties of this work.*

2. This paper is conceived and organized around some mathematical calculations. The risk is that the reader could perceive it as a "synthetic" experiment. I think it is important for NHESS readers to better put their minds on the specific case of study and it could be useful to add a brief description of the study case. This should include a brief description of the test area features and of the debris flows at hand. Also, some more information on the input data are needed (e.g. source of rainfall and debris flows datasets).
*We agree. To make the manuscript more clear and tailored to the NHESS reader, in the revised version of the paper, we will insert a new section describing the study area and the data used in the analysis.*

*A specific section regarding the study area and data (Sect. 3) has been*

*inserted in the revised manuscript.*

3. Discussions are almost missing, mainly they are mixed with the conclusions. I suggest providing separate sections. Discussions should contain an interpretation of the results, while in the conclusions you should summarize the lessons learnt.

*In the revised manuscript, we will divide discussion and conclusions. In the Discussion we will provide a more detailed analysis and interpretation of the results. We will also better highlight what are the characteristics of the event (e.g. the hyetograph shape) that, combined with the inputs uncertainty, mostly affect the variability in the rainfall conditions estimate (i.e. intensity I and duration D). We will stress the impacts on the threshold estimate due to the uncertainty of the input parameters, discussing the resulting variability of the threshold parameters. Finally, we will better highlight the differences between the outcomes obtained performing the standard calibration (i.e. the reference values) and the mean values of the Monte Carlo simulations outputs, both in term of rainfall conditions and threshold.*

*In the Conclusions, we will better highlight the resulting robustness of the BDA method. We will also stress that this robustness could be dependent on the events characteristics used in the estimate. Indeed, applying the BDA method to a different study area, if many of the considered events are characterised by the features that tend to increase the outputs variability (we will describe them in the Discussion), the robustness of the estimate can be undermined. In this case, utmost care in estimating the input parameters is recommended.*

*We will also stress the possibility to apply the developed method for the uncertainty analysis both to different study area (using the BDA) and other thresholds whose calibration is based on physical parameters, as mentioned in a previous answer.*

*Finally, we will emphasize the importance of a validation analysis, not dealt with in this manuscript, to assess the forecast capability of a threshold, once proved to be robust.*

*As explained in point 2 of the major issues of Referee #1, Conclusions and Discussion have been divided. In particular, the discussion has been expanded and included in the results (Sect. 5) while Conclusions have been completely rewritten.*

In particular, some points in my opinion are not clear enough: how does this study help us in predicting stony debris flows?

*Perhaps we were not clear enough in explaining the purpose of this work and actually, some sentences, regarding the forecast skills, were perhaps a bit misleading. The uncertainty analysis aims to quantify the effects of the input uncertainty on the output in order to check the robustness of a method. Instead, to asses the forecast capability of a threshold is the purpose of a validation analysis. This kind of analysis requires a completely different approach than the robustness analysis. This work aims to study the robustness of the BDA approach to calibrate the threshold and, therefore, it does not allow to say anything about the forecast capability of the threshold. We will better clarify these aspects in the Introduction of the*

*revised manuscript.*

*We have clarified the purpose of this work in the Introduction (Lines $59 - 65$) and the misleading sentences have been deleted or revised.*

Does it prove that BDA is robust, or does it prove that the utmost care should be put in calibrating/measuring the input parameters required?
*The results prove that the approach used to estimate the threshold is robust. Indeed, even if some events are characterised by high variability in the rainfall condition (namely in the duration $D$ and/or intensity $I$) due to the uncertainty of the inputs, the resulting threshold has low variability. However, as described in a previous answers, it is advisable to put utmost care in the estimate of the input parameters in some cases, depending on the considered events characteristics. This concept will be inserted and explained in the Conclusions of th revised manuscript.*

*As described in point 2 of the major issues of Referee #1, in the Conclusions of the revised manuscript, we have clarified what are the key findings of the study and their implications and usefulness.*

How does the uncertainty is reflected in the forecasting effectiveness of the resulting threshold? The latter point, in particular, is very important and some tests about that should be shown in the revised version of the manuscript.
*We agree with the referee that the validation of a calibrated threshold is crucial to prove its forecasting effectiveness and to make this tool operational. We are currently working on the validation, developing a suitable method that will be the object of a forthcoming manuscript. Nevertheless, as said before, validation is not the aim of this paper.*
*We preferred to perform a robustness analysis of the calibration method before performing a validation analysis. In our opinion, it would have been useless to validate a threshold obtained from a method later proved not robust.*
*We will better clarify the aim of this work in the Introduction of the revised manuscript and we will stress the importance of the validation in the Conclusions highlighting that this analysis will be the object of a future work.*

*As explained in point 5 of the major issues of Referee #1, in the Conclusions, we have briefly explained that the results of the uncertainty analysis are not useful to assess the forecasting effectiveness of the threshold and that the uncertainty analysis is a prerequisite for the validation analysis (Lines $397 - 401$).*

**Specific comments**

- L17. About review casualties, I suggest to add also the review "Dowling, C. A., & Santi, P. M. (2014). Debris flows and their toll on human life: a global analysis of debris-flow fatalities from 1950 to 2011. Natural hazards, 71(1), 203-227."
  *We thank the referee for the suggestion. We will insert this reference.*

  *The reference has been added (Line $15$).*

- **L23** I suggest "Usually, rainfall thresholds for debris flows initiation are power laws that link the rainfall duration to the rainfall cumulated or Intensity". This is because when dealing with other landslide typologies, other parameters are often used (e.g. antecedent rainfall indexes).
  *We will fix the sentence during the revision.*

  *We have rephrase the sentence according to the review on the rainfall threshold proposed by Segoni et. al. (2018) (Lines $19-22$). We have avoided specifically referring to debris flow to keep the first part of the Introduction more generic.*

- **L25** I would add also the work of Caine (1980), who started this methodology of analysis. Also, I would add some review. And I would substitute the work by Guzzetti by his work published the year after, reviewing ID thresholds (thus, more strictly related to your research).
  *We agree. Other references concerning the rainfall thresholds will be inserted in the revised manuscript and the work by Guzzetti will be substituted by his following work.*

  *Throughout all the Introduction, we have introduced new references concerning the rainfall threshold and the subsequent work by Guzzetti.*

- **L59** The Backward Dynamical Approach (BDA)...
  *We will follow the suggestion.*

  *To be concise, we have only used the acronym BDA already defined in the Introduction.*

- **L106:** Since the study area is located in the Alps, this equation seems very low. Has this threshold been validated before (e.g. in Rosatti et al., 2019)? Could you report the validation result? Before going on with the reading, the readers should know how reliable this threshold is.
  *In Rosatti et al., 2019, two thresholds have been calibrated for the study area following two different approaches to estimate the rainfall condition of the analysed debris flow events: the Critical Duration Method (CMD) (e.g. Restrepo-Posada and Eagleson, 1982; Bonta and Rao, 1988) and the Backward Dynamical Approach (BDA). The obtained CDM-based threshold is the following:*

$$I = 4.91D^{-0.7} \tag{1}$$

*and most of the events have durations in the interval $[1\ h,\ 10\ h]$. This threshold is consistent with the ones obtained by Marra et al., 2014 and by Idanza et al., 2016 for a comparable study area. The BDA-based threshold:*

$$I = 6.2D^{-0.67} \tag{2}$$

*is higher than the CDM-based one at least by $25\%$ (the variation changes based on duration) and most of the durations related to the events belong to the interval $[0.1\ h,\ 1\ h]$. These durations are coherent with the time scale of the observed debris flow.*
*As mentioned in a previous answer, we are currently working on the validation of the threshold.*

How I said, this value seems very low to me and maybe another information should be provided about how the rainfall threshold is operated. E.g. within a long rainfall event a shorter but more intense burst of rain could easily reach the hourly peak intensity of 6.2 or higher. In that case is the threshold exceeded or not?

*According to the BDA method, the intensity and duration associated to an occurred debris flow event are computed starting from the surveyed deposited volume. Then, the rainfall volume per unit area $E$, Eq. (4) of the manuscript, has to be identified in the event hyetograph. If the example described by the referee refers to an occurred event, the $I - D$ couple has to be computed starting from the deposited volume and the other input parameters. If the example refers to the forecasting phase, the referee has highlighted one of the most problematic aspect of the threshold use in the forecast: how to define the exceeding conditions of the threshold if the rainfall strictly pertaining to a debris flow (not all rainfall) and its duration is not known a priori? As mentioned above, we are currently developing a suitable method to validate and use the threshold in the forecast.*

- L112:"some": which one? The ones in tb. 1?
  *Exactly. We will make the sentence more clear.*

  *In the revised manuscript, we have partially rewritten the Method to make the uncertainty analysis and the related uncertain parameters and data clearer.*

- L157-160. This is not clear to me. If I understood correctly, in section 3.1 you set N=100, obtaining 100 random "points" in the I/D plane. Now, you randomly pick one of the I/D points, and you do it 5000 times. I guess most of the points are sampled many times. Because 5000 » 100. Moreover I do not understand how you can generate a threshold for each one of the selected points (5000 thresholds), since you should use many points to define a threshold (in short: many ID couples are needed to define a single a-b couple). I think I misunderstood something in this part, therefore I suggest to rephrase or to explain better.
  *Perhaps we were not clear enough in explaining this part of the method. Performing the first Monte Carlo (MC) simulation (explained in Sect. 3.1), we have obtained $N = 100$ $(I, D)$ couples for each occurred debris flow event. In the case study, we have considered 84 events. This means that the total amount of $(I, D)$ couples obtained from the first MC are $100 \cdot 84 = 8400$. As regards the second MC (described in Sect. 3.2), an input sample is generated randomly selecting one of the possible 100 $(I, D)$ couple for each event. Hence, one sample consists of 84 $(I, D)$ couples. Following this procedure, among the possible combinations, 5000 samples, each comprising 84 $(I, D)$ couples, have been generated. We will rephrase this part of the method to make it clearer.*

  *The generation of the samples of the second MC simulation has been better explained in Sect. 4.3.*

- L175. For what concerns = regarding?
  *We will follow the suggestion.*

  *The sentence has been fixed (Line 332).*

- L209: please check: the flow of the text is broken by the table and the image. It is hard to follow (similar issues elsewhere)
  *We agree. We will optimize the position of the figures and tables.*

  *The position of the figures and tables have been revised.*

- L238-239: please check the text: possible issues.
  *We will fix these sentences inserting a better description of figure 8.*

  *We have reviewed the description of the mentioned figure (Fig. 11 in the revised manuscript) to make it clearer.*

**Graphical improvements**

- Figure 2: I strongly suggest adding another panel to this figure, where the DTM of the study area could be shown together with other relevant data (e.g. debris flows locations).
  *We will follow the suggestion.*

  *The DTM of the study area with the location of the debris flows and the radar has been added in Fig. 2.*

- Please check the text immediately before and after images and tables. Sometimes sentences are split.
  *We agree. We will optimize the position of the figures and tables so as not to split sentences.*

  *In the revised manuscript, we have fixed these graphical issues.*

*Bibliography*

*Bonta, J.V., Rao, A.R., 1988. Factors affecting the identification of independent storm events. J. Hydrol.98, 275–293.*

*Iadanza, C., Trigila, A., Napolitano, F., 2016. Identification and characterization of rain-fall events responsible for triggering of debris flows and shallow landslides. J. Hydrol.541, 230–245.*

*Marra, F., Nikolopoulos, E.I., Creutin, J.D., Borga, M., 2014. Radar rainfall estimation for the identification of debris-flow occurrence thresholds. J. Hydrol.519, 1607–1619.*

*Restrepo-Posada, P.J., Eagleson, P.S., 1982. Identification of independent rainstorms. J. Hydrol.55, 303–319.*

*Zhang, S. J., Xu, C. X., Wei, F. Q., Hu, K. H., Xu, H., Zhao, L. Q., and Zhang, G. P.: A physics-based model to derive rainfall intensity-duration threshold for debris flow. Geomorphology, 351, 106930, 2020.*

---

## Author Response (AR2)

Dear Editor,

we appreciate the suggestions provided. We have modified the manuscript simplifying the structure by merging some sub-sections both in sections 4 and 5. We have also fixed some sentences.
Regarding the discussion, we have preferred to keep it incorporated in the results to avoid having the presentation, the analysis and the interpretation of the results spread out in the paper.

Sincerely,

Marta Martinengo on behalf of all authors